# Self-reported emotional and behavioral problems among school-going adolescents in Nepal—A cross-sectional study

Sirjana Adhikari[1,2]*, Jasmine Ma[2], Suraj Shakya[3], Per Håkan Brøndbo[1], Bjørn Helge Handegård[4], Anne Cecilie Javo[5]

1 Department of Psychology, Faculty of Health Sciences, UiT, The Arctic University of Norway, Tromsø, Norway, 2 Child & Adolescent Psychiatry Unit, Kanti Children's Hospital, Kathmandu, Nepal, 3 Department of Psychiatry and Mental Health, Institute of Medicine, Tribhuvan University, Kathmandu, Nepal, 4 Regional Centre for Child and Youth Mental Health and Child Welfare—North, Faculty of Health Sciences, UiT, The Arctic University of Norway, Tromsø, Norway, 5 Sami National Competence Center for Mental Health (SANKS), Sami Klinihkka, Finnmark Hospital Trust, Karasjok, Norway

* adhikarisirjana16@gmail.com

## Abstract

### Background

Studies on self-reported emotional and behavioral problems (EBPs) among adolescents are still sparse in many low- and middle-income countries. In Nepal, no such studies have been performed on a larger scale, and little is known about self-reported EBPs in the adolescent population.

### Methods

This cross-sectional, school-based study on EBPs included 1904 adolescents aged 11–18 years, enrolled in government and private schools located in 16 districts in Nepal. The Nepali version of the Youth Self Report form was used to assess self-reported EBPs, and the Teacher Report Form was used to assess academic performance. Analysis of variance (ANOVA) was used for gender comparisons on adolescents' EBPs and on academic competence. Multiple regression analysis was done to explore correlates of self-reported EBPs.

### Results

The overall prevalence of self-reported EBPs was 14.2%; 15.6% in boys and 12.9% in girls. The mean Total Problems score was 39.27 (standard deviation = 24.16); no gender differences were observed. Boys scored higher on Externalizing Problems and girls scored higher on Internalizing Problems. The effect sizes for gender comparisons were small with Hedges' g ranging from -0.29 to 0.28. Physical illness and negative/traumatic life events were positively correlated with self-reported EBPs, whereas academic performance was negatively correlated. However, the effect sizes were small ($\eta^2 < 0.02$).

### Conclusion

This study helps to narrow the knowledge gap on the prevalence, magnitude, and types of self-reported EBPs in Nepali adolescents. It demonstrated an association between self-

**Data Availability Statement:** All relevant data are within the paper and its Supporting Information files.

**Funding:** This study led by Sirjana Adhikari is funded by the Norwegian Partnership Program for Global Academic Cooperation (NORPART) 2018/10039 project ("Collaboration in Higher Education in Mental Health between Nepal and Norway") and Child Workers In Nepal (CWIN)-Nepal. The NORPART project funded for the entire research work and the CWIN-Nepal funded for the salary of the principal investigator. URL: 1. NORPART: https://diku.no/en/programmes/norpart-norwegian-partnership-programme-for-global-academic-cooperation 2. CWIN-Nepal: https://www.cwin.org.np/ The funders had no role in study design, data collection and analysis, decision to publish, or preparation of the manuscript.

**Competing interests:** The authors have declared that no competing interests exist.

reported EBPs and academic performance and linked self-reported EBPs to other factors such as negative/traumatic life events and physical illness. The findings might assist health authorities in the planning of mental health services and may also provide valuable background information to clinicians dealing with adolescent mental health problems.

## Introduction

According to the World Health Organization, there are 1.2 billion adolescents worldwide, 90% of whom live in low- and middle-income countries (LMICs) [1]. Generally, LMICs have a younger population than high-income countries; in Nepal, adolescents make up about 20% of the total population [2]. Adolescence is a transition period between childhood and adulthood which is characterized by many physical, psychological, and behavioral changes and adaptation processes. These changes can be emotionally distressing and affect personal well-being, productivity, and quality of life [3, 4].

Globally, one in seven 10-19-year-olds experiences a mental disorder, accounting for 13% of the global burden of disease in this age group [1]. A recent systematic review that examined common mental disorders in adolescents using the General Health Questionnaire (GHQ-12), reported a global prevalence of common mental disorders in adolescents of 25% and 31%, using the GHQ-12 cut-off points 4 and 3, respectively [5]. Further, it has been found that about half of all lifetimes mental health problems start during adolescence [6].

Mental health problems in children and adolescents include several types of emotional and behavioral disorders, i.e., disruptive disorders, depression, anxiety, and pervasive developmental disorders characterized as either "internalizing" (such as depression and anxiety) or "externalizing" (disruptive behaviors such as conduct disorders and attention-deficit/hyperactivity disorder). Emotional and behavioral problems (EBPs) with their related disorders have significant negative impacts on the individual, the family, and society. Despite the relevance of these problems as a leading cause of disability, their long-lasting effects throughout life and impact on the surroundings, the mental health needs of adolescents are often neglected. This is especially true in LMICs, possibly due to limited financial resources and a lack of trained mental health professionals [7, 8]. If adolescent mental health is not addressed, ill health may extend into adulthood, limiting people's opportunities to lead fulfilling lives. Hence, adolescent mental health is a growing public health concern worldwide and should be of particular concern to LMICs [1].

Although the prevalence of mental health disorders in adolescence may be high globally, the prevalence varies across countries and cultures. One reason is that prevalence estimates may vary according to the measurement tools that are used in different studies. Screening instruments that measure EBPs in children and adolescents are commonly used to indicate the prevalence of mental health disorders. For example, scores on the Achenbach System of Empirically Based Assessments (ASEBA) instruments are significantly associated with DSM-5 diagnostic categories, and ASEBA scores can be directly related to DSM-5 diagnostic categories by using the DSM-oriented scales for scoring ASEBA forms[9]. Generally, screening instruments may yield a higher prevalence compared to diagnostic tools [10, 11].

Studies on adolescents' self-reported EBPs are still sparse, especially from LMICs. Few scientific studies have been published on self-reported mental health problems among adolescents in Nepal, except for some smaller studies from different schools and/or from particular geographic areas [12]. The studies reported different prevalences (15%– 30%) and suggested that Nepali adolescents had more internalizing problems than externalizing problems [13, 14].

Recently, the Nepal Health Research Council (NHRC) conducted a national mental health survey with 5888 participants using the "Mini International Neuropsychiatric Interview" as an instrument. This survey yielded a prevalence of mental disorders in adolescents aged 13–17 years of 5.2% [15]. In 2021, a larger epidemiological study of parent-reported EBPs in Nepali children aged 6–18 years was published. This study used the Child Behavior Checklist (CBCL) as an assessment instrument and reported a prevalence of parent-reported EBPs of 18.3% [16].

Age and gender have been found to influence EBPs in studies from around the world. Internalizing problems tend to increase with age, whereas externalizing problems tend to decrease [17, 18]. Across cultures, studies show that girls tend to score higher than boys on internalizing problems, whereas boys tend to score higher on externalizing problems [17, 19–22]. However, more studies from LMICs are warranted to confirm whether these countries follow the same patterns as more developed countries. Further, it has been shown that the magnitude of EBPs and the relative magnitude of internalizing problems versus externalizing problems in adolescents may vary across countries and ethnic groups, suggesting that the impact of culture should be considered when assessing EBPs, especially in the less examined LMICs [18, 23].

Studies have demonstrated a positive relationship between negative/traumatic life events and mental health problems [24–28]. The prevalence of negative life events differs across countries, living environments, and social contexts and may partially explain the cross-cultural variation found in adolescents' EBPs.

Physical illness, including chronic medical conditions that require long-term treatment, can increase the risk of EBPs [29, 30]. Young people with such illnesses are more likely to have higher levels of both internalizing and externalizing problems [31–36]. This may be due to a perceived lack of control over the illness and its symptoms, restrictions to participate in positive activities, and side effects of treatments. Additionally, symptoms of physical illness, such as pain, may lead to elevated problem scores [37].

Factors that are found to be negatively associated with EBPs include child competencies such as academic performance. Studies have reported a positive relationship between mental health and academic achievement [38], whereas mental health problems have been found to have a negative impact [39, 40]. Students with poorer academic performance have reported more EBPs [4, 41].

The present study is part of a larger epidemiological study published on EBPs in Nepali schoolchildren aged 6–18 years [16]. The aims of the present study were to assess the prevalence, magnitude, and types of self-reported EBPs in Nepali adolescents aged 11–18 years and to explore possible gender and age differences. As family correlates of child EBPs had already been examined [42], the present study further aimed to explore the associations between self-reported EBPs and adolescents' factors such as academic performance, physical illness, and the impact of negative/traumatic life events.

## Materials and methods

### Participants and procedure

The present study included 1904 adolescents aged 11–18 years who were attending school during the time of data collection. Sixteen districts: three districts from the Mountains region, seven districts from the Middle hills region, including the Kathmandu district, and six districts from the Terai region were purposively selected. Next, four schools (two governmental and two private) from each district were selected. The purposive sampling technique was chosen for cost-effectiveness and for ease of data collection and travel. Six students (three boys and three girls) from each grade level (grades 6–10) were then randomly selected using random number tables, irrespective of their caste/ethnicity or other background information. The overall participation rate was 99.2%. Missing data was < 0.5% for all instruments.

Before commencing the study, ethical approval was obtained from the Ethical Review Board of the Nepal Health Research Council (NHRC). Project representatives first met with the school management. An invitation letter was then sent to the parents to inform them about the project. Written informed consent was obtained from the parents, and verbal consent was obtained from all participating adolescents. The procedure was according to the NHRC's guidelines at the time. Adolescents completed the Nepali version of the Youth Self Report (YSR) in their classroom. Teachers of the participating adolescents completed the Nepali version of the Teacher's Report Form (TRF). Mothers filled in a background questionnaire. In the case of illiterate parents, research assistants helped to complete the questionnaire. Data were collected during 2017 and 2018 by a team of trained research assistants, monitored by the project leader. Plotting of data was done manually in 2018 by three research assistants, monitored by the project leader.

## Measures

The Nepali versions of the YSR and TRF were made in connection with a previous Nepali Ph. D. dissertation [43]. Written permission to use the Nepali versions was granted by the copyright owner. Both instruments are part of the Achenbach System of Empirically Based Assessment (ASEBA) [44]. They are empirically based and are widely used to assess EBPs in children and adolescents. They have been developed through decades of research and practical experience worldwide to identify actual patterns of child functioning [45]. Confirmatory factor analyses have supported their syndrome structures in dozens of societies [46].

## Youth Self Report

The YSR was used to assess self-reported EBPs in adolescents. It is targeted to the age group 11–18 years and consists of 112 problem items that describe a broad range of problem behaviors [44]. It was modeled after the CBCL which collects parent reports of EBPs and has a similar format only that the items are worded in the first person [37]. The YSR is a reliable instrument with international generalizability and good psychometric properties [44]. It has been found to have good content validity and criterion validity suggesting its ability to discriminate between clinical and non-clinical populations [44]. It can provide guidelines for the use of clinical cut-off points for various purposes. The Nepali version of the YSR was found to be suitable for use in a former Nepali study with good internal consistency [13]. When completing the questionnaire, adolescents were asked to score each item that describes their behavior by circling 0 (not true), 1 (somewhat or sometimes true), or 2 (very true or often true). Problems were scored on a Total Problems scale, two broad-band scales (Internalizing and Externalizing Problems), and eight syndrome scales: Anxious/Depressed, Withdrawn/Depressed, Somatic Complaints, Social Problems, Thought Problems, Attention Problems, Rule-Breaking Behavior, and Aggressive Behavior. The syndrome scales of Anxious/Depressed, Withdrawn/Depressed, and Somatic Complaints are combined to form the Internalizing scale, whereas Rule-Breaking Behavior and Aggressive Behavior are combined to form the Externalizing scale. The Social Problems, Attention Problems, and Thought Problems scales do not belong to either of the broadband scales but are included in the Total Problems scale.

## Teacher's Report Form

The TRF is designed for schoolteachers of children aged 6–18 years and collects information on academic performance, social competencies, adaptive behaviors, and behavior problems. There are 10 items that assess child competencies, and 120 problem items that assess EBPs.

The TRF provides scores of the child's performances in academic subjects on a 5-point Likert scale, from 1 (far below grade) to 5 (far above grade) [44]. In the present paper, we used teacher-reported information on the adolescents' academic performance which was measured across various academic subjects. Each academic subject was rated as 1 (far below grade), 2 (somewhat below grade), 3 (at grade level), 4 (somewhat above grade), and 5 (far above grade).

## Background questionnaire

A background questionnaire was completed by the mothers. It enquired about the child's age and gender; the geographical region of residence; living area (urban, rural, etc.); type of school; caste/ethnicity; parent's occupation, parent's educational level; parent's employment status; family structure and size; family health, including physical illness in the adolescent; family conflicts; child-rearing; and negative/traumatic life events affecting the adolescent. Physical illness was assessed by the question: "Does the child have any chronic physical illness or disabilities?" If yes, the parents were further asked about what type of physical illness (open-ended question). The presence of negative/traumatic life events was assessed by the question: "Has the child experienced any serious life events or trauma during the past 12 months that might have affected him/her psychologically?" If yes, the parents were further asked about what type of negative/traumatic life events (open-ended question).

## Statistical analysis

SPSS statistics version 26.0 for Windows was used for all data analyses. To assess the reliability of the YSR, Cronbach's alphas and item-total correlations were computed. Analysis of variance was used to assess group differences for continuous variables. Hedges' g was computed to indicate effect sizes when comparing two groups. Multiple regression analyses were done to explore correlates of self-reported EBPs. The first model included the following control variables: area of residence, type of school, caste/ethnicity, and parental education. Adolescent factors: age, gender, academic performance, negative/traumatic life events, and chronic physical illness/disabilities were then added to the first model to determine differences in the explained variances ($R^2$). Partial eta squared was used to measure the effect sizes. We used a significance level of 0.01 for all tests.

## Results

### Reliability analysis of the Nepali version of the Youth Self Report

Cronbach's alpha for the YSR was found to be within the acceptable range, with values above 0.7 for all syndrome scales except the Withdrawn/Depressed and Social Problems scales, which had a somewhat lower level. We also calculated the correlations between each syndrome scale and each of its subitems, which revealed weak to strong associations according to Cohen's classification [47, 48]. For the Anxious/Depressed scale, items like "fears at school", "perfectionism", and "thoughts about suicide" were weakly correlated, whereas items such as "nervous or tense", "afraid of certain animals, situations or places", and "worries" were strongly correlated. For the Withdrawn/Depressed scale, the items "enjoys little" and "withdrawn" had the weakest correlation, and "sadness" had the strongest correlation. For Somatic Complaints, the item "eye problem" was moderately correlated, and "nausea" was strongly correlated. For Rule-Breaking Behavior, "use of alcohol" had the weakest correlation, and the item related to "lies and cheating" had the highest correlation, though it was still moderate. For Aggressive Behavior, "demanding attention" had the weakest correlation, and the item related to "stubbornness" had the strongest correlation. For Thought Problems, "harms self" had the weakest

**Table 1. Internal consistency of the Youth Self Report (YSR) for Nepali adolescents (N = 1904).**

| YSR syndrome scales | Cronbach's alpha α | Pearson's correlation ($r_{min}$ -$r_{max}$) |
| --- | --- | --- |
| Anxious/Depressed (13 items) | 0.76 | 0.200–0.550 |
| Withdrawn/Depressed (8 items) | 0.68 | 0.317–0.497 |
| Somatic Complaints (10 items) | 0.78 | 0.308–0.548 |
| Social Problems (11 items) | 0.69 | 0.203–0.390 |
| Thought Problems (12 items) | 0.73 | 0.166–0.504 |
| Attention Problems (9 items) | 0.73 | 0.299–0.504 |
| Rule-Breaking Behavior (15 items) | 0.71 | 0.235–0.441 |
| Aggressive Behavior (17 items) | 0.84 | 0.254–0.571 |

correlation, and "having strange ideas" had the strongest correlation. For Attention Problems, "acting young for his/her age" had the weakest correlation, and "inattentiveness" was most strongly correlated. For Social Problems, "preference for younger children" had the weakest correlation, and other items were moderately correlated (Table 1).

## Background information

The mean age of the adolescents was 14.5 years (standard deviation, SD = 1.78). Most of the participants were from the Middle hills and Terai regions and from semi-urban areas. Father's occupation was mostly farming, and most mothers were housewives. A considerable number of fathers were migrant workers (16%). Most fathers had a secondary school education (53%), whereas mothers had a lower level of education. About 11% of the adolescents had at least one illiterate parent. About half of the participants belonged to the higher Hindu caste groups, whereas 36% were from ethnic/ indigenous minorities. Less than 10% of the adolescents had been affected by negative/traumatic life events in the past 12 months, and most of them (89%) had no reported chronic physical illness/disabilities (Table 2).

## Prevalence of emotional and behavioral problems

The prevalence of self-reported EBPs was calculated according to the YSR manual, i.e., based on American cut-off points indicating "normal", "borderline" and "clinical" range of problems [44]. Prevalence of Total Problems was found to be 14.2%; 15.6% for boys and 12.9% for girls. Relatively more adolescents scored in the clinical range on Internalizing Problems (24.0%) than on Externalizing Problems (7.4%) (Table 3).

**Table 2. Distribution of selected background variables (N = 1904).**

| Selected background variables | N (%) |
| --- | --- |
| **Gender** | |
| Boys | 951 (49.9) |
| Girls | 953 (50.1) |
| **Geographic region of residence** | |
| Mountain region | 235 (12.3) |
| Middle hills region (incl. the capital Kathmandu) | 964 (50.6) |
| Terai region | 705 (37.0) |
| **Area of residence** | |

*(Continued)*

**Table 2.** (Continued)

| Selected background variables | N (%) |
|---|---|
| Rural | 422 (22.2) |
| Semiurban | 1080 (56.7) |
| Urban | 402 (21.1) |
| **Type of school** | |
| Government | 954 (50.1) |
| Private | 949 (49.9) |
| **Caste/ethnicity** | |
| Hindu high caste groups[1] | 1011 (53.1) |
| Indigenous groups / ethnic minorities[2] | 685 (36.0) |
| Hindu low caste groups[3] | 168 (8.8) |
| Others[4] | 40 (2.1) |
| **Parental education[5]** | |
| Illiterate | 204 (10.7) |
| Primary level[6] | 409 (21.5) |
| Secondary level[7] | 1082 (56.8) |
| University level[8] | 209 (11.0) |
| **Father's occupation** | |
| Public service | 231 (12.1) |
| Private business | 463 (24.3) |
| Farmer | 602 (31.6) |
| Migrant worker | 298 (15.7) |
| Others | 310 (16.3) |
| **Mother's occupation** | |
| Housewife | 1302 (68.4) |
| Public service | 82 (4.3) |
| Private business | 201 (10.6) |
| Farmer | 228 (12.0) |
| Migrant worker | 33 (1.7) |
| Others | 58 (3.0) |
| **Chronic physical illness/disabilities** | |
| Yes | 201 (10.6) |
| No | 1702 (89.4) |
| **Negative/traumatic life events in the past 12 months** | |
| Yes | 139 (7.3) |
| No | 1764 (92.7) |

1: Hindu High caste: Caste-origin Hill group, Madhesi caste-origin groups (Socio-economic level 1), and Madhesi caste-origin groups (Socio-economic level 2)

2: Indigenous groups / ethnic minorities (Hill Aadibasi/Janajati groups and Madhesi (Terai) Aadibasi/Janajati groups)

3: Hindu Low caste groups (Hill Low caste or Dalits and Madhesi Low caste groups (socio-economic level 3)

4: Others: Musalman and others

5: In the households with two parents, the higher education level was used

6: Primary level education consists of 5 years of education from grade 1 to 5.

7: Secondary level education consists of 12 years of education up to grade 12.

8: University level education includes 13 years of education and above

**Table 3. Prevalence of emotional and behavioral problems in Nepali adolescents by gender.**

| Domain | Gender | | Total (N = 1904) |
|---|---|---|---|
| | Boys (n = 951) | Girls (n = 953) | |
| **Total Problems (T-score) *** | | | |
| Normal (< 60) | 70.9% | 77.6% | 74.3% |
| Borderline (60–63) | 13.6% | 9.4% | 11.5% |
| Clinical (> 63) | 15.6% | 12.9% | 14.2% |
| **Internalizing Problems (T-score) **** | | | |
| Normal (< 60) | 56.3% | 65.3% | 60.8% |
| Borderline (60–63) | 15.0% | 15.4% | 15.2% |
| Clinical (> 63) | 28.7% | 19.3% | 24.0% |
| **Externalizing Problems (T-score)** | | | |
| Normal (< 60) | 84.4% | 86.4% | 85.4% |
| Borderline (60–63) | 6.6% | 7.9% | 7.2% |
| Clinical (> 63) | 8.9% | 5.8% | 7.4% |
| **Anxious/Depressed (T-score) **** | | | |
| Normal (< 65) | 79.7% | 87.1% | 83.4% |
| Borderline (65–69) | 14.3% | 8.3% | 11.3% |
| Clinical (> 69) | 6.0% | 4.6% | 5.3% |
| **Withdrawn/Depressed (T-score)** | | | |
| Normal (< 65) | 84.3% | 87.8% | 86.1% |
| Borderline (65–69) | 11.7% | 8.2% | 9.9% |
| Clinical (> 69) | 4.0% | 4.0% | 4.0% |
| **Somatic Complaints (T- score) **** | | | |
| Normal (< 65) | 81.3% | 78.4% | 79.9% |
| Borderline (65–69) | 6.0% | 16.5% | 11.3% |
| Clinical (> 69) | 12.6% | 5.1% | 8.9% |
| **Social Problems (T- score)** | | | |
| Normal (< 65) | 79.3% | 80.0% | 79.7% |
| Borderline (65–69) | 12.2% | 10.4% | 11.3% |
| Clinical (> 69) | 8.4% | 9.6% | 9.0% |
| **Thought Problems (T- score)** | | | |
| Normal (< 65) | 91.3% | 93.7% | 92.5% |
| Borderline (65–69) | 5.3% | 4.4% | 4.8% |
| Clinical (> 69) | 3.5% | 1.9% | 2.7% |
| **Attention Problems (T- score)** | | | |
| Normal (< 65) | 94.0% | 92.8% | 93.4% |
| Borderline (65–69) | 3.8% | 4.6% | 4.2% |
| Clinical (> 69) | 2.2% | 2.6% | 2.4% |
| **Aggressive Behavior (T- score)** | | | |
| Normal (< 65) | 89.8% | 91.5% | 90.6% |
| Borderline (65–69) | 5.9% | 5.7% | 5.8% |
| Clinical (> 69) | 4.3% | 2.8% | 3.6% |
| **Rule-Breaking Behavior (T- score)** | | | |
| Normal (< 65) | 95.7% | 97.4% | 96.5% |
| Borderline (65–69) | 3.1% | 2.1% | 2.6% |

(*Continued*)

**Table 3.** (Continued)

| Domain | Gender | | Total (N = 1904) |
|---|---|---|---|
| | Boys (n = 951) | Girls (n = 953) | |
| Clinical (> 69) | 1.3% | 0.5% | 0.9% |

*P < 0.01

**P < 0.001

For gender comparisons, the Pearson's chi square test was used.

## Magnitude of emotional and behavioral problems by gender in Nepali adolescents

Boys scored significantly higher than girls on Externalizing Problems and Rule-Breaking Behavior. Girls scored significantly higher than boys on Internalizing Problems, Anxious/Depressed, and Somatic Complaints. However, there was no gender difference in the magnitude of Total Problems. The Hedges' g ranged from −0.29 to 0.28, indicating small effect sizes for gender comparisons according to Cohen [47] (Table 4).

## Mean item scores

The highest mean item scores were found for items that fall into the following syndrome scales: Anxious/Depressed, Withdrawn/Depressed, Somatic Complaints, Social Problems, Thought Problems, and Rule-Breaking Behavior (Table 5).

## Academic performance by gender

The mean score of academic performance was 2.99 (SD = 1.00) for the total sample, 2.94 (SD = 1.01) for boys, and 3.03 (SD = 0.98) for girls. The gender difference was not significant. Hedges' g was 0.09, suggesting a small effect size for gender comparison according to Cohen [47].

## Correlates of self-reported emotional and behavioral problems

For Total Problems, the variance accounted by the control variables was 1.7% ($R^2 = 0.017$). The variance increased to 3.7% ($R^2 = 0.037$) after adding the adolescent factors. For

**Table 4. Youth Self Report mean scale scores in Nepali adolescents by gender (N = 1904).**

| Problems | Boys (n = 951) Mean (SD) | Girls (n = 953) Mean (SD) | Total (n = 1904) Mean (SD) | F | Effect Size (Hedges' g) |
|---|---|---|---|---|---|
| Total Problems | 39.02 (23.19) | 39.52 (25.10) | 39.27 (24.16) | 0.209 | −0.02 |
| Externalizing Problems | 9.17 (7.23) | 7.96 (6.87) | 8.57 (7.07) | 14.08** | 0.17 |
| Internalizing Problems | 13.21 (7.89) | 15.30 (9.44) | 14.26 (8.76) | 27.20** | −0.24 |
| Anxious/Depressed | 5.58 (3.59) | 6.55 (4.25) | 6.07 (3.96) | 28.64** | −0.24 |
| Withdrawn/Depressed | 3.87 (2.61) | 4.00 (2.87) | 3.93 (2.75) | 1.03 | −0.04 |
| Somatic Complaints | 3.75 (3.11) | 4.74 (3.65) | 4.25 (3.43) | 40.18** | −0.29 |
| Social Problems | 4.92 (3.18) | 4.81 (3.27) | 4.86 (3.22) | 0.58 | 0.03 |
| Thought Problems | 3.99 (3.21) | 3.70 (3.29) | 3.84 (3.26) | 3.76 | 0.08 |
| Attention Problems | 4.26 (3.09) | 4.19 (3.21) | 4.22 (3.15) | 0.22 | 0.02 |
| Rule-Breaking Behavior | 3.22 (2.99) | 2.42 (2.64) | 2.82 (2.85) | 38.35** | 0.28 |
| Aggressive Behaviors | 5.95 (4.84) | 5.54 (4.89) | 5.74 (4.87) | 3.39 | 0.08 |

*P < 0.01

**P < 0.001

SD: standard deviation.

**Table 5. Items with the highest mean scores in Nepali adolescents (N = 1904).**

| Items with the highest mean scores* | Syndrome scales |
|---|---|
| 32. Have to be perfect | Anxious/Depressed |
| 11. Too dependent on adults | Social Problems |
| 31. Fears of doing bad | Anxious/Depressed |
| 5. Enjoys little | Withdrawn/Depressed |
| 36. Accidently gets hurt a lot (clumsy) | Social Problems |
| 63. Prefers older kids than kids of same age | Rule-Breaking Behavior |
| 47. Has nightmares | Somatic Complaints |
| 56 b. Headaches | Somatic Complaints |
| 76. Sleeps less than most kids | Thought Problems |
| 29. Fears (afraid of certain animals, situations, or places other than school) | Anxious/Depressed |

*Items are listed in descending order of mean scores

Internalizing Problems, the variance accounted by the control variables was 2.2% ($R^2$ = 0.022), which increased to 5.1% ($R^2$ = 0.051) after adding the adolescent factors. For Externalizing Problems, the variance increased from 1.5% ($R^2$ = 0.015) to 3.8% ($R^2$ = 0.038) after adding the adolescent factors. When all the control variables and the adolescent factors were simultaneously entered into the model, the child's age was positively correlated with Externalizing Problems. The female gender was strongly associated with Internalizing Problems, i.e., girls scored higher than boys, whereas the male gender was more strongly associated with Externalizing Problems, i.e., boys scored higher than girls. Further, chronic physical illness/disabilities and child experience of negative/traumatic life events were positively correlated with Total Problems, Internalizing Problems, and Externalizing Problems. Academic performance was negatively correlated with Total Problems. However, all effect sizes were small, with partial eta squares less than 0.02 (Table 6).

## Discussion

The study assessed the prevalence, magnitude, and type of self-reported EBPs among school-going adolescents in 16 different districts of Nepal. In addition, correlates of self-reported EBPs, such as age, gender, academic performance, chronic physical illness/disabilities, and negative/traumatic life events, were explored.

Earlier, different norm groups (high, medium, and low) were constructed for the YSR. These were based on the omnicultural mean of 36.67 (SD = 6.23) that was found in a large, epidemiological study by averaging the Total Problems scores of 34 cultures [21]. Later, Nepal was placed in the highest rank based on results from a former study of adolescents residing in the Western development region of Nepal [13]. However, the present study showed a mean Total Problems score of 39.27 (SD = 24.16) which suggests that Nepal should be placed among the medium-scoring countries since the mean score lied within one SD of the omnicultural mean. The difference in mean scores between the two Nepali studies might be related to methodological issues such as smaller sample size and limited geographic area in the earlier study [13].

In the present study, the prevalence of self-reported EBPs among Nepali adolescents was 14.2%. This prevalence is higher than the mean prevalence of self-reported mental health problems for adolescents in LMICs described in an earlier review [49]. It should be noted that the higher prevalence might be due to methodological issues. Studies using screening instruments are likely to overestimate the prevalence rates compared to studies using validated diagnostic

**Table 6. Correlates of self-reported emotional and behavioral problems in Nepali adolescents.**

| Background variables | Total Problems | | | Internalizing Problems | | | Externalizing Problems | | |
|---|---|---|---|---|---|---|---|---|---|
| | Regression coefficient (B) (SE) | F | Partial$\eta^2$ | Regression coefficient (B) (SE) | F | Partial$\eta^2$ | Regression coefficient (B) (SE) | F | Partial $\eta^2$ |
| *Control variables* | | | | | | | | | |
| **Living area** (Reference group: Urban) | | 9.37** | 0.01 | | 13.17** | 0.01 | | 4.14* | 0.00 |
| Rural | 7.43 (1.74) | 18.23** | 0.01 | 3.19 (.63) | 25.91** | 0.01 | 1.37 (0.51) | 7.24* | 0.00 |
| Semi-urban | 4.73 (1.44) | 10.69** | 0.01 | 1.94 (.52) | 13.98** | 0.01 | 1.05 (0.42) | 6.20* | 0.00 |
| **Type of school** (Reference group: private school) | −4.61 (1.23) | 13.98** | 0.01 | −1.06 (.44) | 5.72* | 0.00 | −1.75 (0.36) | 23.58** | 0.01 |
| **Parental education** (Reference group: University level) | | 0.93 | 0.00 | | 1.56 | 0.00 | | 0.77 | 0.00 |
| Illiterate | 1.87 (2.51) | 0.54 | 0.00 | 0.98 (0.91) | 1.17 | 0.00 | 0.45 (0.74) | 0.37 | 0.00 |
| Primary level | 3.48 (2.24) | 2.43 | 0.00 | 1.56 (0.81) | 3.76 | 0.00 | 0.93 (0.65) | 2.02 | 0.00 |
| Secondary level | 2.63 (1.87) | 1.99 | 0.00 | 1.34 (0.67) | 3.96 | 0.00 | 0.66 (0.55) | 1.46 | 0.00 |
| **Caste/ethnicity** (Reference group: Hindu high caste group) | | 2.30 | 0.00 | | 2.39 | 0.00 | | 1.60 | 0.00 |
| Indigenous groups/ethnic minorities | 2.36 (1.21) | 3.80 | 0.00 | 0.74 (0.44) | 2.86 | 0.00 | 0.70 (0.35) | 3.96 | 0.00 |
| Hindu low caste groups | 4.04 (2.04) | 3.92 | 0.00 | 1.52 (0.73) | 4.28 | 0.00 | 0.71 (0.60) | 1.41 | 0.00 |
| Others | 4.83 (3.87) | 1.56 | 0.00 | 2.14 (1.39) | 2.34 | 0.00 | 1.06 (1.13) | 0.88 | 0.00 |
| *Adolescent factors* | | | | | | | | | |
| **Age** | 0.59 (0.31) | 3.64 | 0.00 | 0.15 (0.11) | 1.72 | 0.00 | 0.25 (0.09) | 7.46* | 0.00 |
| **Gender** (Reference group: Boys) | 1.02 (1.10) | 0.87 | 0.00 | 2.22 (0.40) | 31.58** | 0.02 | −1.05 (0.32) | 10.78** | 0.01 |
| **Physical illness** (Reference group: No physical illness) | 5.31 (1.81) | 8.61* | 0.01 | 2.15 (0.65) | 10.89** | 0.01 | 1.22 (0.53) | 5.31 | 0.003 |
| **Negative/traumatic life events in past 12 months** (Reference group: No negative life events) | 8.11 (2.15) | 14.25** | 0.01 | 2.56 (0.77) | 10.95** | 0.01 | 2.04 (0.63) | 10.53** | 0.01 |
| **Academic performance** | −1.38 (0.56) | 5.99* | 0.00 | −0.31 (0.20) | 2.26 | 0.00 | −0.33 (0.17) | 3.99 | 0.00 |
| **Full model** | $R^2 = 0.037$ | | | $R^2 = 0.051$ | | | $R^2 = 0.038$ | | |
| **For control variables only** | $R^2 = 0.017$ | | | $R^2 = 0.022$ | | | $R^2 = 0.015$ | | |

F = "F-test statistic"; SE = standard error; $\eta^2$: eta squared

*P < 0.01

**P < 0.001

interviews or rates based on diagnostic criteria [10, 50]. However, our finding is within the range of prevalence found in some previous Nepali studies [13, 14, 51–55].

As in other studies that used the YSR in South Asia [13, 14, 20], the prevalence of internalizing problems was much higher than that of externalizing problems (24.0% vs. 7.4%). This finding has also been reported in studies from other parts of the world, e.g., Europe [19] and Africa [56]. Further, our finding is in line with a recent American study suggesting a trend of increasing internalizing symptoms and decreasing externalizing symptoms among adolescents during the last decennium [57]. The finding of higher internalizing problems relative to externalizing problems in countries like Nepal could also be explained by the cultural sanction and cultural facilitation model described by Weisz et al. [58]. "Cultural facilitation" refers to a cluster of behaviors encouraged by a culture, whereas "cultural sanction" refers to a cluster of behaviors that are verbally disapproved, negatively sanctioned, or punished by that culture. Asian cultures tend to deemphasize verbal expression of aggression, acts of physical aggression, and rule-breaking behavior in children and adolescents (i.e., cultural sanction) which

might facilitate the occurrence of internalizing problems [59]. Additionally, the value placed on family connectedness in Asian cultures is associated with a greater tolerance for Internalizing Problems (i.e., cultural facilitation) and a lower tolerance for Externalizing Problems [60]. Other studies have demonstrated that ratings of EBPs differ between countries. For example, in a cross-cultural study, Japanese adolescents reported a lower rate of externalizing behaviors (Rule-Breaking and Aggressive Behaviors) than adolescents from Greece, Russia, and Sweden [61]. However, cultural differences might be only one of several reasons for the differences in internalizing versus externalizing problems. Biological factors such as genes and sex hormones also impact the expression of symptoms, especially in adolescence [62, 63]. For instance, the male hormone, testosterone, is found to be associated with aggressive, rule-breaking behaviors [64]. However, to examine the biological aspect of adolescent behaviors was beyond the scope of the present study.

In our study, the prevalence of internalizing problems in boys was 28.7% compared to 19.3% in girls. This higher prevalence for boys should be regarded with caution as the prevalence estimates were based on American norms, and we cannot conclude that Nepali children's problems were categorized according to relevant cut-offs. When we examined the mean scores (raw scores), girls had a higher mean score of Internalizing Problems than boys (15.30 versus 13.21). Our findings of gender differences in the magnitude (means) of self-reported EBPs converge with findings from a comprehensive, international meta-analytic review [65]; and with findings from other international studies [13, 17–20, 66]. Previous studies have suggested that girls' tendency to internalize problems is mostly related to their daily hassles and the parent-child relationship [67], and to gender-specific coping patterns, such as girls' increased tendency to ruminate when compared to boys [68]. The same explanations might apply to Nepali girls, along with the impact of other, additional factors such as cultural norms. For example, Nepali boys and girls are expected to behave differently. Girls are expected to engage in household chores, communicate through expressions of fearfulness, and remain silent. To protect girls' marriage prospects, local cultural practices provide girls with less cultural space to behave in a disruptive manner [69]. Our finding that boys had higher scores on Rule-Breaking Behavior than girls is consistent with a previous Nepali study [13]. A possible explanation could be that boys spend more time outside of the home and are expected to show their masculinity by confronting fearful situations while limiting their emotional expression [69]. More studies on gender-related patterns in adolescents' EBPs are warranted across cultures, especially from less investigated LMICs like Nepal.

We found that Externalizing Problems seemed to rise with increasing age, whereas Total Problems and Internalizing Problems did not. This finding is consistent with a previous study [19], but contrasts with most of the international literature which has suggested that older adolescents report more Total Problems and Internalizing Problems [18, 21] and less Aggressive Behavior [17]. Our finding might be due to a lack of culturally appropriate norms with which to compare the types of problems. Owing to the small effect sizes, these finding needs replication, and future studies are needed to verify the results.

The single items with the highest mean scores in our study, were self-criticism, fearfulness, inability to experience pleasure, too much dependency on adults, sleep problems, and headaches. These are somewhat different from the items that were most frequently reported in a previous international study that compared ratings of EBPs on the YSR in general population samples from 24 different countries [18]. However, one item, item no 32: "Have to be perfect", was found to have a high mean score in both studies. Most of the frequently reported items in our study were related to somatic problems and to anxiety and depressive problems which might be due to the cultural expression of distress in Nepali culture. Items related to the use of drugs such as alcohol or tobacco, were among the least reported, which might be due to

underreporting. Indeed, adolescents might want to keep any drug use a secret, given the taboo of substance use in Nepali society [70].

In accordance with other studies, negative/traumatic life events were positively associated with EBPs [25, 28, 51, 71]. The reason for increased EBPs among adolescents suffering from traumatic life events could be related to the stress and difficulty of coping with such events. In LMICs, like in Nepal, this association needs to be further explored considering the challenging socio-political contexts, the poverty and economic adversities, and the frequency of natural disasters that many children and adolescents in these countries must endure, and which might make them more vulnerable to mental health problems.

An important risk factor for mental health problems during adolescence is the presence of chronic physical illness/disabilities [72]. Consistent with the international literature [29–33], we found a positive association between physical illness and self-reported EBPs in Nepali adolescents. The fact that adolescents with chronic physical illness/disabilities are more likely to suffer from EBPs implicates the necessity to offer mental health assessments and treatment to adolescents with physical illnesses.

Further, we found that academic performance was negatively correlated with self-reported EBPs in our Nepali adolescents. The finding is consistent with the international literature [4, 38–40]. It seems that EBPs in adolescents not only cause pain and distress but also negatively influence their academic performance [73]. However, we cannot conclude whether EBPs lead to poor performance or vice-versa. It may even be a self-reinforcing circle. It should be mentioned that in Nepal, adolescents with poor academic performance are likely to be treated unfavorably and physically punished both at home and school, as this may be considered the best way to discipline a child and help him/her perform better [74]. This response may lead the adolescent to develop feelings of guilt, fear, violation, loss of control, and low self-esteem, which in turn could lead to more dysfunctional behavior. One explanation for the negative association between academic performance and self-reported EBPs might be the specific negative effect of attention problems on academic performance [41, 75]. Regardless of the explanation, early identification of adolescents' EBPs and timely intervention are recommended to improve academic attainment.

## Strengths and limitations

The present study is a large-scale epidemiological study on EBPs among school-going Nepali adolescents based on their self-reports, covering several districts in different parts of the country and all three main geographic regions. Sound methodology and thorough procedures were used in data collection, which resulted in a very high participation rate (99%). However, we cannot claim that the results are representative of the whole country, as districts were not randomly sampled.

The instruments used in this study, the YSR and the TRF, are internationally well-established instruments and have been shown to be valid in many different cultures. In our study, Cronbach's alpha displayed overall good internal consistency for the YSR ($> 0.7$). However, some of the syndrome scales (i.e., the Withdrawn/Depressed scale and the Social Problem scale) had a lower Cronbach's alpha suggesting inconsistencies in responses on different items within the scales which might be due to a lack of culturally appropriate items. Low scores indicated that the set of items did not reliably measure the same construct. Another reason for varying internal consistency might have been the varying number of items in different scales. The Withdrawn/Depressed scale has eight indicators, and this is the lowest number of items of the eight YSR syndrome scales. A full validation study of the YSR in a Nepali context is warranted and should be addressed in future investigations.

It should be noted that our prevalence estimates were based on American norms of the YSR as Nepali norms are lacking. Hopefully, future studies will provide separate norms for Nepal. As child and adolescent mental health services are gradually being developed in Nepal, this kind of research should be more feasible in the future.

Although the YSR includes diverse single problem items, additional, more culturally appropriate items might have given different results. As for Somatic Complaints, our study suggested that the YSR did not include some culturally relevant items in a Nepali context. When asked to describe their somatic symptoms, adolescents mostly reported headache, abdominal pain, and nausea, all of which are listed in the YSR, but they also frequently reported dizziness and fainting which are not specific options in the YSR. Future studies are warranted to explore in more detail what culturally specific somatic symptoms are reported by Nepali adolescents.

The present study was a school-based study. It likely missed some drop-out students who might have had more EBPs than their school-going peers [76]. Hence, their inclusion might have increased both the prevalence and magnitude of EBPs. However, in contemporary Nepali society, few students (about 5%) leave school every year [77]. Reasons for drop-outs might be poverty, child marriage, and child labor, along with school-related problems such as corporal punishment, academic difficulties, difficulty accessing school, and gender disparities [77]. In future Nepali studies, an additional examination of drop-out students is recommended to get a more accurate estimate of self-reported EBPs in adolescents.

Another limitation of our study was the rather unspecified items pertaining to physical illness and traumatic life events. It would have strengthened the study if we had been able to examine the impact of the different types of chronic physical illnesses/disabilities on EBPs. However, to explore the specific impact of each condition was beyond the scope of our study. Further, the reliability of parent-reported traumatic events might be questionable since many adolescents may not disclose some traumatic experiences such as sexual abuse.

## Implications for mental health services

The study might inform policymakers in Nepal about the need for adolescent mental health services, prevention work, and guide in service planning. It may also make practitioners more aware of the magnitude of EBPs among school-going adolescents and of the different correlates that may impact their mental health conditions.

## Conclusion

This study focuses on the prevalence, magnitude, and type of self-reported EBPs among school-going adolescents in Nepal and explores possible correlates. It helps to narrow the knowledge gap in the existing literature about adolescents' EBPs in LMICs and demonstrates that in a country like Nepal, many adolescents are likely to suffer from various types of mental health problems that need attention. Further, it highlights the importance of adolescent factors that might be associated with EBPs. Given the scarcity of data on adolescents' EBPs in Nepal, our study might provide useful information to researchers, clinicians, and policymakers alike and may be particularly useful in preventive mental health work.

## Supporting information

**S1 File.**
(SAV)

## Acknowledgments

We are grateful to all participating Nepali adolescents and their teachers, and to the team of data enumerators and supervisors for making this study possible. Further, we would like to extend our gratitude to Dr. Arun Raj Kunwar and his child and adolescent psychiatry team at Kanti Children's Hospital, Kathmandu, for their support.

## Author Contributions

**Conceptualization:** Sirjana Adhikari, Anne Cecilie Javo.

**Formal analysis:** Sirjana Adhikari.

**Methodology:** Jasmine Ma, Anne Cecilie Javo.

**Project administration:** Jasmine Ma.

**Supervision:** Suraj Shakya, Per Håkan Brøndbo, Bjørn Helge Handegård, Anne Cecilie Javo.

**Writing – original draft:** Sirjana Adhikari.

**Writing – review & editing:** Sirjana Adhikari, Jasmine Ma, Suraj Shakya, Per Håkan Brøndbo, Bjørn Helge Handegård, Anne Cecilie Javo.

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
