## [Decision Letter · Decision Letter 0]

21 Dec 2022

PONE-D-22-03677Self-reported emotional and behavior problems in Nepali adolescents - a general population-based studyPLOS ONE

Dear Dr. Adhikari,

Thank you for submitting your manuscript to PLOS ONE. I sincerely apologise for the unusually delayed review timeframe. Your manuscript has been assessed by one reviewer, whose comments are appended below. After careful consideration, we feel that it has merit but does not fully meet PLOS ONE’s publication criteria as it currently stands. Therefore, we invite you to submit a revised version of the manuscript that addresses the points raised during the review process. Please note that we have only been able to secure a single reviewer to assess your manuscript. We are issuing a decision on your manuscript at this point to prevent further delays in the evaluation of your manuscript. Please be aware that the editor who handles your revised manuscript might find it necessary to invite additional reviewers to assess this work once the revised manuscript is submitted. However, we will aim to proceed on the basis of this single review if possible.

We look forward to receiving your revised manuscript.

Kind regards,

Emily Chenette

Editor in Chief

PLOS ONE

Journal Requirements:

“We would like to acknowledge the NORPART 2018/10039 project (“Collaboration in Higher Education in Mental Health between Nepal and Norway”) and CWIN -Nepal for supporting this study. The charges for online publication of this paper have been funded by a grant from the publication fund of UiT the Arctic University of Norway. We are grateful to all participating Nepali adolescents and their teachers and to the team of data enumerators and supervisors for making this study possible. Further, we would like to extend our gratitude to Dr. Arun Raj Kunwar and the child and adolescent psychiatry team at Kanti Children’s Hospital, Kathmandu, for their support. “

“This study led by Sirjana Adhikari is funded by the Norwegian Partnership Program for Global Academic Cooperation (NORPART) 2018/10039 project (“Collaboration in Higher Education in Mental Health between Nepal and Norway”) and Child Workers In Nepal (CWIN)-Nepal. The NORPART project funded for the entire research work and the CWIN-Nepal funded for the salary of the principal investigator.

URL:

1. NORPART : https://diku.no/en/programmes/norpart-norwegian-partnership-programme-for-global-academic-cooperation

2. CWIN-Nepal: https://www.cwin.org.np/

Reviewers' comments:

Reviewer's Responses to Questions

**Comments to the Author**

1. Is the manuscript technically sound, and do the data support the conclusions?

Reviewer #1: Partly

2. Has the statistical analysis been performed appropriately and rigorously? 

Reviewer #1: Yes

3. Have the authors made all data underlying the findings in their manuscript fully available?

Reviewer #1: Yes

4. Is the manuscript presented in an intelligible fashion and written in standard English?

Reviewer #1: Yes

5. Review Comments to the Author

Reviewer #1: The manuscript presents a report of findings on emotional and behavioral problems based on a school based survey of adolescents in Nepal. The population coverage and the relatively large sample size are very good. The authors have presented the magnitude of EBP and the factors associated with it. Although the report has some merit, there are some issues the authors need to address to improve its scientific value.

1. The introductory section cites several prevalence estimates of EBP in the youth. In this section, it is important to present the validity of the measures used with respect to clinical diagnoses. Self reports are general used as screening tool and generally tend to overestimate prevalence. This is important because the authors conclude that the results may be relevant for clinicians.

2. While the data presented in Table 3 shows that the clinical degree of EBP based on T-scores is higher among boys than girls for the Total, Internalizing and Externalizing categories, the authors rather chose to discuss only the results in Table 4. It would be more appropriate to discuss the findings in Table 3 or provide a good reason why the authors chose not to.

3. The discussion section emphasizes too much the cultural aspects of why externalizing symptoms are lower in prevalence. This is probably only one of the several reasons. Biological factors such as sex hormones especially among adolescents play an important role in regard to aggressive behaviors. The discussion should include the potential role of biological factors in the expression of symptoms from earlier publications.

6. PLOS authors have the option to publish the peer review history of their article (what does this mean?). If published, this will include your full peer review and any attached files.

Reviewer #1: No

---

## [Author Response · Author response to Decision Letter 0]

1 Feb 2023

Response to the Reviewer’s comments

Dear Reviewer, 

We would like to thank you for reviewing our manuscript. Your comments and questions have been valuable to improve the quality of our paper. We have revised the manuscript according to your advice, to the best of our knowledge. References have been revised accordingly, and we have added some more references in the discussion section. According to the advice of the journal, we have provided a thorough language proof-reading of the whole manuscript to improve the language so that it will now be clear, correct, and unambiguous, and typographical and grammatical errors have been corrected. Below, we have given our answers to your comments one by one. In doing so, we have referred to the corresponding lines in our revised and proof-read manuscript with track changes (see Revised manuscript with track changes). 

Reviewer comment 1:

“The introductory section cites several prevalence estimates of EBP in the youth. In this section, it is important to present the validity of the measures used with respect to clinical diagnoses. Self- reports are general used as screening tool and generally tend to overestimate prevalence. This is important because the authors conclude that the results may be relevant for clinicians”.

Our Reply: 

Thank you for this comment. The introduction was written to give an overview of prevalence estimates of adolescents’ mental health problems in general populations as measured by screening instruments as this was the focus of our paper. However, we agree that prevalence obtained by screening instruments often differ from those obtained by diagnostic tools or -interviews. Hence, we have added the following sentences in the “Introduction” section: 

“One reason is that the prevalence may vary according to the measurement tools that are used in different studies. Generally, screening instruments may yield higher prevalence compared to diagnostic tools [10, 11].” See line number 72-74 in the revised manuscript. 

In addition, we have added the following sentences about the validity of the tool that we have used (The Youth Self-Report – YSR) in the “Materials and Methods” section: 

“It has been found to have good content validity and criterion validity suggesting its ability to discriminate between clinical and non-clinical populations[53]. It can provide guidelines for the use of clinical cut-off points for various purposes.” See line number 170-173 in the revised manuscript. 

Finally, we added the following sentences in the “Discussion” section as follows: 

“It should be noted that the higher prevalence in our study might be due to methodological issues since we used a screening instrument (YSR). Studies using screening tools are likely to overestimate the prevalence rates compared to studies using validated diagnostic interviews or rates based on diagnostic criteria [10, 62]. See line number 330-333 in the revised manuscript.

Reviewer comment 2: 

“While the data presented in Table 3 shows that the clinical degree of EBP based on T-scores is higher among boys than girls for the Total, Internalizing and Externalizing categories, the authors rather chose to discuss only the results in Table 4. It would be more appropriate to discuss the findings in Table 3 or provide a good reason why the authors chose not to”.

Our Reply: 

Thank you for the comment. In Table 3, we calculated the prevalence estimates based on American norms (T-scores) and in Table 4, we presented the mean scores (raw scores). As the T-scores were based on American norms, we cannot say for sure that Nepali children’s problems were categorized according to relevant cut-offs. As we regarded the raw scores as more valid due to the unavailability of Nepali norms for the YSR, we decided to discuss our results mostly based on Table 4 and not to pay too much attention to the “clinical status” of the children. Further, by using the scale scores, we could keep information about each individual’s level of problems, while the classification into “normal”, “borderline” and “clinical” would make us lose some information. Lost information is yet another reason for not discussing the prevalence results into much detail. To clarify, we have now extended the discussion about internalizing versus externalizing problems in the “Discussion” section of the revised manuscript by adding the following explanation: 

“In our study, the prevalence of internalizing problems in boys was 28.7% compared to 19.3% for girls. This higher prevalence for boys should be regarded with caution as the prevalence estimates were based on American norms, and we cannot say for sure that Nepali children’s problems were categorized according to relevant cut-offs. Hence, we will focus more on differences in mean scores (raw scores) that are independent of norms.” See line number 358-362 in the «Discussion» section of the revised manuscript. 

We also extended the paragraph about internalizing versus externalizing problems in the “Discussion” section by adding the following about a recent American study: 

“Further, our findings are in line with a recent American study suggesting a trend of increasing internalizing symptoms and decreasing externalizing symptoms among adolescents during the last decennium [64]. See line number 338-340 in the revised manuscript. 

Reviewer comment 3: 

“The discussion section emphasizes too much the cultural aspects of why externalizing symptoms are lower in prevalence. This is probably only one of the several reasons. Biological factors such as sex hormones especially among adolescents play an important role in regard to aggressive behaviors. The discussion should include the potential role of biological factors in the expression of symptoms from earlier publications”.

Our Reply: 

Thank you for this important input. We agree that we should have mentioned other explanations of behavior, such as biological factors. According to your advice, we have now added the potential role of biological factors and referred to earlier publications in the “Discussion” section:

”However, cultural differences might be only one of several reasons for the differences in internalizing versus externalizing problems. Biological factors such as genes and sex hormones also impact the expression of symptoms, especially in adolescence [69, 70]. For instance, the male hormone, testosterone, is found to be associated with aggressive, rule breaking behaviors[71]. However, to examine the biological aspect of adolescent behaviors was beyond the scope of the present study”. See line number 352-357 in the revised manuscript.

---

## [Decision Letter · Decision Letter 1]

12 Apr 2023

PONE-D-22-03677R1Self-reported emotional and behavioral problems in Nepali adolescents - a general population-based studyPLOS ONE

Dear Dr. Adhikari,

Thank you for submitting your manuscript to PLOS ONE. After careful consideration, we feel that it has merit but does not fully meet PLOS ONE’s publication criteria as it currently stands. Therefore, we invite you to submit a revised version of the manuscript that addresses the points raised during the review process.

We look forward to receiving your revised manuscript.

Kind regards,

Yubaraj Adhikari, PhD

Academic Editor

Plos One

Journal Requirements:

**Additional Editor Comments:**

Dear Ms. Adhikari,

This is to inform you that your article - PONE-D-22-03677 - "Self-reported emotional and behavioral problems in Nepali adolescents - a general population-based study - Revision 1" went through a peer review process.

Plos One team acknowledges the efforts of the authors and reviewers to review and enhance the quality of this research paper. Thanks to the authors for their valuable work. Plos One editorial team and reviewers concluded that the revised manuscript incorporated many concerns raised during the review process of the original submission. The quality of the manuscript has improved. However, this manuscript requires addressing some of the issues in relation to language, referencing, writing style, and formulation of conclusions. The reviewers have underlined the following major concerns.

The revised manuscript still requires the correction of typos and grammatical errors. The writing style and presentation of the findings need revisions before its final publication process.

The ethical and funding-related issues concerning this research paper are now adequately addressed in the revised manuscript. Authors are asked to rework the comments and suggestions of the reviewers and resubmit the manuscript for the next steps.

You can find a summary of comments presented by the reviewers. Please review them, address the concern of the reviewers, and submit the revised manuscript with appropriate explanations to the reviewers stating details on addressing their comments.

Comments to the Authors

Reviewer #2

1. If the authors have adequately addressed your comments raised in a previous round of review and you feel that this manuscript is now acceptable for publication, you may indicate that here to bypass the “Comments to the Author” section, enter your conflict of interest statement in the “Confidential to Editor” section, and submit your "Accept" recommendation.

Accept. All comments have been addressed.

2. Is the manuscript technically sound, and do the data support the conclusions?

Yes

3. Has the statistical analysis been performed appropriately and rigorously?

I don’t know

4. Have the authors made all data underlying the findings in their manuscript fully available?

Yes

5. Is the manuscript presented in an intelligible fashion and written in standard English?

Yes

6. Review Comments to the Author

The reviewed version has addressed all the previous comments. The article is presented in a comprehensible way, which is easy to follow.

7. PLOS authors have the option to publish the peer review history of their article (what does this mean?). If published, this will include your full peer review and any attached files. Do you want your identity to be public for this peer review? For information about this choice, including consent withdrawal, please see our Privacy Policy.

Yes: Bhava Poudyal

Reviewer #3:

1. If the authors have adequately addressed your comments raised in a previous round of review and you feel that this manuscript is now acceptable for publication, you may indicate that here to bypass the “Comments to the Author” section, enter your conflict of interest statement in the “Confidential to Editor” section, and submit your "Accept" recommendation.

Accept. All comments have been addressed

2. Is the manuscript technically sound, and do the data support the conclusions?

Yes

3. Has the statistical analysis been performed appropriately and rigorously?

Yes

4. Have the authors made all data underlying the findings in their manuscript fully available?

Yes

5. Is the manuscript presented in an intelligible fashion and written in standard English?

Yes

6. Review Comments to the Author

Not Available

7. PLOS authors have the option to publish the peer review history of their article (what does this mean?). If published, this will include your full peer review and any attached files. Do you want your identity to be public for this peer review? For information about this choice, including consent withdrawal, please see our Privacy Policy.

Yes: Professor Dr. AJAY RISAL

Reviewer #4

1. If the authors have adequately addressed your comments raised in a previous round of review and you feel that this manuscript is now acceptable for publication, you may indicate that here to bypass the “Comments to the Author” section, enter your conflict of interest statement in the “Confidential to Editor” section, and submit your "Accept" recommendation.

Minor Revision. All comments have been addressed

2. Is the manuscript technically sound, and do the data support the conclusions?

Yes

3. Has the statistical analysis been performed appropriately and rigorously?

Yes

4. Have the authors made all data underlying the findings in their manuscript fully available?

Yes

5. Is the manuscript presented in an intelligible fashion and written in standard English?

No

6. Review Comments to the Author

I think this paper is an important addition to the child and adolescent psychiatry literature. Congratulations to all authors! However, I would like to recommend a few changes in all parts of the manuscript in terms of grammar/language, conciseness in writing, and the addition of a few pieces of information.

I. Though the author has mentioned thorough proofreading of language being carried out in response to the previous reviewer’s comments there are still typographical and grammar mistakes in the manuscript which need correction.

eg:

Line 120 “child competences”-should have been competencies.

In line 173 “TRF”-should have been “YSR”

There are still grammatical errors that need to be addressed like needs the addition of ‘the’; ‘for’; ‘a’; the need to change the word ‘from’ to ‘in’, and the correct plural/singular forms.

II. Regarding conciseness in writing for clear understanding;

• Lines 75-76 need revision;

• Lines 116-117 need correction;

• Lines 136-137 need to be re-written;

• Line 183 needs to be rewritten

• Lines 205-206 need revision,

• Lines 315-316 need revision.

III. Addition of information:

Introduction 1st paragraph needs elaboration on EBP and an explanation of how EBP is related to the mental health of adolescents. The manuscript seems to lack this point creating confusion between EBP and mental health problems as these terms are independently written.

There is not much background information regarding adaptive functioning however, items related to adaptive functioning have been explored. Similarly, it has not been discussed much in the discussion section questioning its pre-requisite.

Is it correct YRF has 105 items and TRF has a total of 128 items which has been mentioned in the manuscript?

The manuscript has not mentioned copyright issues of both YRF and TRF. Neither has it been mentioned in the manuscript by Ma and colleagues.

The description of YRF and TRF are very imprecise. What are the items related to adaptive functioning within TRF?

Were the physical illnesses and life events assessed by only one question or there were further questions? Like duration for chronic illness and disabilities.

What was the result of chronic illnesses and disabilities in terms of types?

What was the result of serious life events in terms of types? Since the question was asked to parents, how do you justify its reliability, especially for traumas like sexual abuse?

Why do you think the internal consistency of some item syndrome scales of YSR and the correlation between some sub-items-syndrome scales was weak?

Line 235-236, how is the mention of the sample population to be in accordance with the overall Nepal population distribution justifiable when your sampling was purposive?

Since YSR was not validated in Nepali and did not have the norms, I am interested to know what could have been the finding of EBP if the norms from a culturally similar country were applied!

Did you explore the relationship between EBP and two broad-band scales? It would have been interesting to know if internalizing problems were common in adolescents suffering from severe traumas.

Lastly, as you have explained the reasons for the increased prevalence of internalizing problems in adolescents our culture of discouraging verbal expression and emotional expression from childhood additionally might have played the role in the occurrence of internalizing problems.

7. PLOS authors have the option to publish the peer review history of their article (what does this mean?). If published, this will include your full peer review and any attached files. Do you want your identity to be public for this peer review? For information about this choice, including consent withdrawal, please see our Privacy Policy.

Yes: No

Reviewer #5

1. If the authors have adequately addressed your comments raised in a previous round of review and you feel that this manuscript is now acceptable for publication, you may indicate that here to bypass the “Comments to the Author” section, enter your conflict of interest statement in the “Confidential to Editor” section, and submit your "Accept" recommendation.

Accept. All comments have been addressed

2. Is the manuscript technically sound, and do the data support the conclusions?

Yes

3. Has the statistical analysis been performed appropriately and rigorously?

Yes

4. Have the authors made all data underlying the findings in their manuscript fully available?

Yes

5. Is the manuscript presented in an intelligible fashion and written in standard English?

Yes

6. Review Comments to the Author

The Abstract clearly describes the study's aims, methods, and results. The title of the article is informative and relevant. Relevant references have been listed.

Introduction and Method:

The introduction and background list what are already known about this topic, and are clear on the research question, which is justified considering what is presently known. Subject selection is clearly stated, and the variables are defined and measured appropriately. The noted study methods are valid and reliable with the provision of enough detail to replicate the study.

Results:

Results are presented appropriately, with the provided text adding further context to the data. It is clear to the reader in terms of findings what is statistically significant and practically meaningful.

Discussion:

Results were discussed well, and placed in context, without being overinterpreted. Conclusions answer the aims of the study and are supported by results and references. The limitations of the study listed are opportunities to inform future research.

Overall:

This journal article aims to explore the prevalence of EBPs and associated adolescent factors among school-going adolescents in Nepal. It contributes to the dearth of information on this topic from an LMIC in the South Asian context and allows for the comparison of Adolescent EBPS in this region. The study design was appropriate to answer the aims, and the results do justice in adding the Nepali adolescent experience of EBPs and filling in missing information on associated factors in the country context as well as providing other relevant data on the topic. The previous reviewers’ comments have been addressed adequately and the article remains consistent with itself. The article is recommended to be accepted for publication.

7. PLOS authors have the option to publish the peer review history of their article (what does this mean?). If published, this will include your full peer review and any attached files. Do you want your identity to be public for this peer review? For information about this choice, including consent withdrawal, please see our Privacy Policy.

No

Reviewer #6

1. If the authors have adequately addressed your comments raised in a previous round of review and you feel that this manuscript is now acceptable for publication, you may indicate that here to bypass the “Comments to the Author” section, enter your conflict of interest statement in the “Confidential to Editor” section, and submit your "Accept" recommendation.

Minor Revision. All comments have been addressed

2. Is the manuscript technically sound, and do the data support the conclusions?

Yes

3. Has the statistical analysis been performed appropriately and rigorously?

Yes

4. Have the authors made all data underlying the findings in their manuscript fully available?

Yes

5. Is the manuscript presented in an intelligible fashion and written in standard English?

Yes

6. Review Comments to the Author

• Was there any feasibility criteria used for the purposive sampling method?

• Please clarify the statement "since assent by the youths themselves was not mandatory at that time"

• In table 1 "geographic region of residence"; in methods authors have mentioned about Kathmandu as a separate entity. Kindly clarify.

• Table 1: Physical illness present or absent is very vague term. what physical illness were included? May be clarified in method section.

• The claim of being the first study in Nepal would be very misleading. There has been national mental health survey in 2020 and data are available on mental health of adolescent as well.

• You may add some important points in discussion like the treatment gap and implication of findings to the practicing psychiatrists.

7. PLOS authors have the option to publish the peer review history of their article (what does this mean?). If published, this will include your full peer review and any attached files. Do you want your identity to be public for this peer review? For information about this choice, including consent withdrawal, please see our Privacy Policy.

Yes, Dr. Pawan Sharma

Reviewer #7

1. If the authors have adequately addressed your comments raised in a previous round of review and you feel that this manuscript is now acceptable for publication, you may indicate that here to bypass the “Comments to the Author” section, enter your conflict of interest statement in the “Confidential to Editor” section, and submit your "Accept" recommendation.

Minor Revision. All comments have been addressed

2. Is the manuscript technically sound, and do the data support the conclusions?

Not Available

3. Has the statistical analysis been performed appropriately and rigorously?

4. Not Available

5. Have the authors made all data underlying the findings in their manuscript fully available?

Yes

6. Is the manuscript presented in an intelligible fashion and written in standard English?

Not Available

7. Review Comments to the Author

Over all comment-

Pertinent, useful, important and well designed study; and well written manuscript, which needs to address a few issues for publication.

Some issues-

• Regarding title, ‘Self-reported emotional and behavioral problems in Nepali adolescents - a general population-based study’, is it

- in Nepali adolescent students?

- A school- based study?

- As discussed by the authors themselves, whether enrolled in community or school may have implication.

• Concern regarding section ‘Introduction’ is only word count/limit, if it is as per journal format? Or may make a succinct introduction from current one.

• Minor language and punctuation errors, e.g. better use past tense than present in ‘Materials and Method’ section (e.g., in line- 193 previous version of manuscript).

• Authors wrote in discussion, ‘Consistent with some studies [24], we found that Externalizing Problems seemed to rise with increasing age.’ (line 401). Please indicate based on which result (table No.?), this has been inferred? I do not see any such result, i.e. distribution as per age groups so that we can infer it.

• What was done for those adolescent students with EBP? (An ethical concern.)

• References need to follow the proper style consistently, e.g. in references 10, 16, 19.

8. PLOS authors have the option to publish the peer review history of their article (what does this mean?). If published, this will include your full peer review and any attached files. Do you want your identity to be public for this peer review? For information about this choice, including consent withdrawal, please see our Privacy Policy.

Yes, Dhana Ratna Shakya

Reviewer #8

1. If the authors have adequately addressed your comments raised in a previous round of review and you feel that this manuscript is now acceptable for publication, you may indicate that here to bypass the “Comments to the Author” section, enter your conflict of interest statement in the “Confidential to Editor” section, and submit your "Accept" recommendation.

Minor Revision.

2. Is the manuscript technically sound, and do the data support the conclusions?

Yes

3. Has the statistical analysis been performed appropriately and rigorously?

I don’t know

4. Have the authors made all data underlying the findings in their manuscript fully available?

Yes

5. Is the manuscript presented in an intelligible fashion and written in standard English?

No

6. Review Comments to the Author

Title:

The title says the study is a general population-based study. The methodology says the study

sample is school attending population. The school-attending population, however, is not the

same as the general population. School students, asylum, jail inmates, etc. are special

population groups that differ from the general population and they do not represent the general

population. I mean adolescents in the general population are not the same as adolescents who

go to school/college. Therefore, the title of this article seems to be misleading.

Abstract:

The first sentence of the conclusion section (of the abstract) is the repetition of what is already

mentioned in the background section (of the abstract).

Introduction:

Order of information can put in better way. Intially part: What is EBPs, why is it important to

know? What does the existing studies say about prevalence and correlates of EBPs and in the

later part what are the lacunae in current research and why you want to do this study?

Materials and methods:

In section ‘Participants and procedure’: I think it is not good to tell readers to search for

another article to see the detailed procedure and methodology of this study. The second

sentence has the word ‘analytical sample’. I do not think ‘analytical sample’ means anything.

Are YSR and TRF Nepali translations are validated?

References:

Referencing styles are not uniform.

Example:

Sahayaraj.R W. Behavioural and emotional problems in school going adolescents.

International Research journal of Management Sociology & Humanities, 2015; 6(12):357-363.

Doi: https://doi.org/10.32804/IRJMSH

In the above referencing, you can omit 3 in 363 to write it like below

Sahayaraj.R W. Behavioural and emotional problems in school going adolescents.

International Research journal of Management Sociology & Humanities, 2015; 6(12):357-63.

Doi: https://doi.org/10.32804/IRJMSH

General comments:

The manuscript is better to be proofread by a professional English editor. For text clarity, we

can refrain from using additional words, mostly filler words/connecting words, which can be

omitted, e.g. "respectively", "thus", "hence", "therefore", "however, "indeed", etc. Long

sentences can be converted into more than one shorter sentence to make them more readable.

There are a few spelling mistakes. Eg Analyses of variance should be spelled correctly as

Analysis of variance (plz see abstract). Hedges’ g should be spelled as Hedges’ G. References

numbers are generally to be put at the end of the sentences. There are a few instances where

reference numbers are put in the middle of the sentences.

7. PLOS authors have the option to publish the peer review history of their article (what does this mean?). If published, this will include your full peer review and any attached files. Do you want your identity to be public for this peer review? For information about this choice, including consent withdrawal, please see our Privacy Policy.

Yes, Khagendra Kafle, MBBS, MD

Plos One requests the author to resubmit the revised manuscript within 30 days so that the publication and review process remains robust for this manuscript.

With warm regards,

Yubaraj Adhikari, PhD

Academic Editor

Plos One

Reviewers' comments:

Reviewer's Responses to Questions

**Comments to the Author**

1. If the authors have adequately addressed your comments raised in a previous round of review and you feel that this manuscript is now acceptable for publication, you may indicate that here to bypass the “Comments to the Author” section, enter your conflict of interest statement in the “Confidential to Editor” section, and submit your "Accept" recommendation.

Reviewer #2: All comments have been addressed

Reviewer #3: All comments have been addressed

Reviewer #4: All comments have been addressed

Reviewer #5: All comments have been addressed

Reviewer #6: All comments have been addressed

Reviewer #7: (No Response)

Reviewer #8: (No Response)

2. Is the manuscript technically sound, and do the data support the conclusions?

Reviewer #2: Yes

Reviewer #3: Yes

Reviewer #4: Yes

Reviewer #5: Yes

Reviewer #6: Yes

Reviewer #7: (No Response)

Reviewer #8: Yes

3. Has the statistical analysis been performed appropriately and rigorously? 

Reviewer #2: I Don't Know

Reviewer #3: Yes

Reviewer #4: Yes

Reviewer #5: Yes

Reviewer #6: Yes

Reviewer #7: Yes

Reviewer #8: I Don't Know

4. Have the authors made all data underlying the findings in their manuscript fully available?

Reviewer #2: Yes

Reviewer #3: Yes

Reviewer #4: Yes

Reviewer #5: Yes

Reviewer #6: Yes

Reviewer #7: (No Response)

Reviewer #8: Yes

5. Is the manuscript presented in an intelligible fashion and written in standard English?

Reviewer #2: Yes

Reviewer #3: Yes

Reviewer #4: No

Reviewer #5: Yes

Reviewer #6: Yes

Reviewer #7: Yes

Reviewer #8: No

6. Review Comments to the Author

Reviewer #2: The reviewed version has addressed all the previous comments. The article is presented in a comprehensible way, which is easy to follow.

Reviewer #3: (No Response)

Reviewer #4: (No Response)

Reviewer #5: Abstract, title, references:

The Abstract clearly describes the study's aims, methods, and results. The title of the article is informative and relevant. Relevant references have been listed.

Introduction and Method:

The introduction and background list what are already known about this topic, and are clear on the research question, which is justified considering what is presently known. Subject selection is clearly stated, and the variables are defined and measured appropriately. The noted study methods are valid and reliable with the provision of enough detail to replicate the study.

Results:

Results are presented appropriately, with the provided text adding further context to the data. It is clear to the reader in terms of findings what is statistically significant and practically meaningful.

Discussion:

Results were discussed well, and placed in context, without being overinterpreted. Conclusions answer the aims of the study and are supported by results and references. The limitations of the study listed are opportunities to inform future research.

Overall:

This journal article aims to explore the prevalence of EBPs and associated adolescent factors among school-going adolescents in Nepal. It contributes to the dearth of information on this topic from an LMIC in the South Asian context and allows for the comparison of Adolescent EBPS in this region. The study design was appropriate to answer the aims, and the results do justice in adding the Nepali adolescent experience of EBPs and filling in missing information on associated factors in the country context as well as providing other relevant data on the topic. The previous reviewers’ comments have been addressed adequately and the article remains consistent with itself. The article is recommended to be accepted for publication.

Reviewer #6: 1. Was there any feasibility criteria used for the purposive sampling method?

2. Please clarify the statement "since assent by the youths themselves was not mandatory at that time"

3. In table 1 "geographic region of residence"; in methods authors have mentioned about Kathmandu as a separate entity. Kindly clarify.

4. Table 1: Physical illness present or absent is very vague term. what physical illness were included? May be clarified in method section.

5. The claim of being the first study in Nepal would be very misleading. There has been national mental health survey in 2020 and data are available on mental health of adolescent as well.

6. You may add some important points in discussion like the treatment gap and implication of findings to the practicing psychiatrists.

Reviewer #7: Over all comment-

Pertinent, useful, important and well designed study; and well written manuscript, which needs to address a few issues for publication.

Some issues-

1. Regarding title, ‘Self-reported emotional and behavioral problems in Nepali adolescents - a general population-based study’, is it

- in Nepali adolescent students?

- A school- based study?

- As discussed by the authors themselves, whether enrolled in community or school may have implication.

2. Concern regarding section ‘Introduction’ is only word count/limit, if it is as per journal format? Or may make a succinct introduction from current one.

3. Minor language and punctuation errors, e.g. better use past tense than present in ‘Materials and Method’ section (e.g., in line- 193 previous version of manuscript).

4. Authors wrote in discussion, ‘Consistent with some studies [24], we found that Externalizing Problems seemed to rise with increasing age.’ (line 401). Please indicate based on which result (table No.?), this has been inferred? I do not see any such result, i.e. distribution as per age groups so that we can infer it.

5. What was done for those adolescent students with EBP? (An ethical concern.)

6. References need to follow the proper style consistently, e.g. in references 10, 16, 19.

Reviewer #8: Title:

The title says the study is a general population-based study. The methodology says the study

sample is school attending population. The school-attending population, however, is not the

same as the general population. School students, asylum, jail inmates, etc. are special

population groups that differ from the general population and they do not represent the general

population. I mean adolescents in the general population are not the same as adolescents who

go to school/college. Therefore, the title of this article seems to be misleading.

Abstract:

The first sentence of the conclusion section (of the abstract) is the repetition of what is already

mentioned in the background section (of the abstract).

Introduction:

Order of information can put in better way. Intially part: What is EBPs, why is it important to

know? What does the existing studies say about prevalence and correlates of EBPs and in the

later part what are the lacunae in current research and why you want to do this study?

Materials and methods:

In section ‘Participants and procedure’: I think it is not good to tell readers to search for

another article to see the detailed procedure and methodology of this study. The second

sentence has the word ‘analytical sample’. I do not think ‘analytical sample’ means anything.

Are YSR and TRF Nepali translations are validated?

References:

Referencing styles are not uniform.

Example:

Sahayaraj.R W. Behavioural and emotional problems in school going adolescents.

International Research journal of Management Sociology & Humanities, 2015; 6(12):357-363.

Doi: https://doi.org/10.32804/IRJMSH

In the above referencing, you can omit 3 in 363 to write it like below

Sahayaraj.R W. Behavioural and emotional problems in school going adolescents.

International Research journal of Management Sociology & Humanities, 2015; 6(12):357-63.

Doi: https://doi.org/10.32804/IRJMSH

General comments:

The manuscript is better to be proofread by a professional English editor. For text clarity, we

can refrain from using additional words, mostly filler words/connecting words, which can be

omitted, e.g. "respectively", "thus", "hence", "therefore", "however, "indeed", etc. Long

sentences can be converted into more than one shorter sentence to make them more readable.

There are a few spelling mistakes. Eg Analyses of variance should be spelled correctly as

Analysis of variance (plz see abstract). Hedges’ g should be spelled as Hedges’ G. References

numbers are generally to be put at the end of the sentences. There are a few instances where

reference numbers are put in the middle of the sentences.

7. PLOS authors have the option to publish the peer review history of their article (what does this mean?). If published, this will include your full peer review and any attached files.

Reviewer #2: **Yes: **Bhava Poudyal

Reviewer #3: **Yes: **Professor Dr. AJAY RISAL

Reviewer #4: No

Reviewer #5: No

Reviewer #6: **Yes: **Dr. Pawan Sharma

Reviewer #7: **Yes: **Dhana Ratna Shakya

Reviewer #8: No

---

## [Author Response · Author response to Decision Letter 1]

17 May 2023

Response to the Reviewers’ comments

Dear Reviewers,

We would like to thank you for reviewing our manuscript. Your comments and questions have been valuable to improve the quality of our paper. We have revised the manuscript according to your advice, to the best of our knowledge. References have been revised accordingly. According to the advice of the journal, we have provided a thorough language proof-reading of the whole manuscript to improve the language so that it will now be clear, correct, and unambiguous, and typographical and grammatical errors have been corrected. Below, we have given our answers to your comments one by one. In doing so, we have referred to the corresponding lines in our revised and proof-read manuscript with track changes (see Revised manuscript with track changes). 

Reviewer #2

Reviewer’s comment:

1. If the authors have adequately addressed your comments raised in a previous round of review and you feel that this manuscript is now acceptable for publication, you may indicate that here to bypass the “Comments to the Author” section, enter your conflict of interest statement in the “Confidential to Editor” section, and submit your "Accept" recommendation.

Accept. All comments have been addressed.

2. Is the manuscript technically sound, and do the data support the conclusions?

Yes

3. Has the statistical analysis been performed appropriately and rigorously?

I don’t know.

4. Have the authors made all data underlying the findings in their manuscript fully available?

Yes

5. Is the manuscript presented in an intelligible fashion and written in standard English?

Yes

6. Review Comments to the Author

The reviewed version has addressed all the previous comments. The article is presented in a comprehensible way, which is easy to follow.

Our reply:

Thank you for the positive comment and for accepting our paper for further processing. We appreciate your time and effort to go through the manuscript. 

Reviewer #3

1. If the authors have adequately addressed your comments raised in a previous round of review and you feel that this manuscript is now acceptable for publication, you may indicate that here to bypass the “Comments to the Author” section, enter your conflict of interest statement in the “Confidential to Editor” section, and submit your "Accept" recommendation.

Accept. All comments have been addressed. 

2. Is the manuscript technically sound, and do the data support the conclusions?

Yes

3. Has the statistical analysis been performed appropriately and rigorously?

Yes

4. Have the authors made all data underlying the findings in their manuscript fully available?

Yes

5. Is the manuscript presented in an intelligible fashion and written in standard English?

Yes

6. Review Comments to the Author

Not Available

Our reply:

Thank you for accepting our paper for further process. We appreciate your time and effort to go through the manuscript. 

Reviewer # 4

Reviewer’s comment 1:

I think this paper is an important addition to the child and adolescent psychiatry literature. Congratulations to all authors! However, I would like to recommend a few changes in all parts of the manuscript in terms of grammar/language, conciseness in writing, and the addition of a few pieces of information. 

I. Though the author has mentioned thorough proofreading of language being carried out in response to the previous reviewer’s comments there are still typographical and grammar mistakes in the manuscript which need correction.eg: Line 120 “child competences”-should have been competencies. In line 173 “TRF”-should have been “YSR”

There are still grammatical errors that need to be addressed like needs the addition of ‘the’; ‘for’; ‘a’; the need to change the word ‘from’ to ‘in’, and the correct plural/singular forms.

Our Reply: 

Thank you for this comment. We have corrected “child competences” to “child competencies” and we have changed “TRF” to “YSR”. Further, we have checked the whole manuscript again for typographical and grammar mistakes and corrected the grammatical errors. 

Reviewer’s comment 2:

Regarding conciseness in writing for clear understanding: lines 75-76 need revision; lines 116-117 need correction; lines 136-137 need to be re-written; line 183 needs to be rewritten; lines 205-206 need revision; lines 315-316 need revision.

Our Reply: 

Thank you for the comment. We have now made revisions for more conciseness and clear understanding. In the revised manuscript, lines 75-76 are written as: “Studies on adolescents’ self-reported EBPs are still sparse, especially from LMICs” (See line 84 in the revised manuscript). Lines 116-117 have been removed to shorten the introduction. Lines 136-137 have been removed and the information about data collection procedures has been given in details in the same paragraph. Line 183 has been rewritten as: “There are 10 items that assess child competencies, and 120 problem items that assess EBPs”(See lines 177-178 in the revised manuscript). Lines 205-206 have now been written as: “Partial eta squared was used to measure the effect sizes” (See lines 204-205 in the revised manuscript). Lines 315-316 have now been removed. 

Reviewer’s comment 3: 

Introduction: 1st paragraph needs elaboration on EBP and an explanation of how EBP is related to the mental health of adolescents. The manuscript seems to lack this point creating confusion between EBP and mental health problems as these terms are independently written.

Our Reply:

Thank you for the comment. We have now revised the introduction to clarify how EBP is related to adolescents’ mental health – see lines 57-68, and 77-82 in the revised manuscript.

Reviewer’s comment 4:

There is not much background information regarding adaptive functioning however, items related to adaptive functioning have been explored. Similarly, it has not been discussed much in the discussion section questioning its pre-requisite.

Our Reply:

Thank you for the comment. You are right that we had not given any information regarding adaptive functioning in the introduction and did not discuss about it in the discussion. Since adaptive functioning was not the focus of this study, the table (i.e., Table 6.) showing gender differences in adaptive functioning has now been removed from the revised manuscript. We have included the reports on academic performance in the text (lines 283-286 in the revised manuscript). Academic performance was addressed both in the introduction and in the discussion. 

Reviewer’s comment 5:

Is it correct YSR has 105 items and TRF has a total of 128 items which has been mentioned in the manuscript?

Our Reply:

The correct number of problem items in the YSR is 112, not 105 problem items as we wrote – thank you for informing us about this mistake. We have corrected it in the revised manuscript in line number 158 in the revised manuscript. Also, we have clarified in the text that the TRF has 120 problem items and 10 additional items describing about the context of the students as well as asking about their academic performance – see lines 175-183 in the revised manuscript. 

Reviewer’s comment 6:

The manuscript has not mentioned copyright issues of both YRF and TRF. Neither has it been mentioned in the manuscript by Ma and colleagues.

Our Reply:

Thank you for rightly pointing this out. We have now added information about copyright permission in the revised manuscript - see the measures section, line numbers 150-151. 

Reviewer’s comment 7:

The description of YRF and TRF are very imprecise. What are the items related to adaptive functioning within TRF?

Our Reply:

Thank you for the comment. We have now added information about the YSR, see lines 170-174 in the revised manuscript. As for adaptive functioning, it was not mentioned in the description of the TRF because it was beyond the scope of our study. The Table 6, in which adaptive functioning was presented, is therefore removed. The item that we had included in the introduction, in the aims of the study, and in analyses, was academic performance, and we have kept that. It is described in the paragraph about the TRF, lines 176-183. 

Reviewer’s comment 8: 

Were the physical illnesses and life events assessed by only one question or there were further questions? Like duration for chronic illness and disabilities. What was the result of chronic illnesses and disabilities in terms of types?

Our Reply:

Thank you for the comment. Additional question about duration was not asked. Physical illness was assessed by the question: “Does the child have any chronic physical illness or disabilities?” If parents answered, “yes”, they were further enquired about the type of physical illness. This additional question was an open-ended question and has now been mentioned in the “Background Questionnaire”, in the Methods section – see lines 190-191 in the revised manuscript. The types of illness reported by the parents included a wide range of conditions. We did not analyze the impact of each condition on EBP as this was beyond the scope of the present study. We have explained this in the “Strengths and limitations” part of the Discussion, see lines 453-456 in the revised manuscript. 

The presence of negative/traumatic life events was assessed by the question: “Has the child experienced any serious life events or trauma during the past 12 months that might have affected him/her psychologically?” Response options were “yes” or “no”. If the response was yes, parents were asked to describe about the types. This additional, open-ended question has now been added as information in the “Background Questionnaire”, see lines 194-195 in the revised manuscript. Types of traumatic life events were described by the parents as e.g., serious accident, domestic violence, death of close family member, parental divorce, absence of close relative or friend, economic crisis or hardships, or other burdensome events or trauma. Due to the various types and the subsequent small numbers of each type, further analyses of their effect on EBPs were not performed.

Reviewer’s comment 9: 

What was the result of serious life events in terms of types? Since the question was asked to parents, how do you justify its reliability, especially for traumas like sexual abuse?

Our Reply:

Thank you for the comment. Types of life events were explained in our reply to comment 8 – see above. None of the parents reported that their adolescent had been sexually abused in the past year. We agree that the reliability of parent-reported trauma pertaining to sexual abuse might be questionable. We have now commented on this in the “Strengths and limitation” part of the Discussion, see lines 456-458 in the revised manuscript.

Reviewer’s comment 10: 

Why do you think the internal consistency of some item syndrome scales of YSR and the correlation between some sub-items-syndrome scales was weak?

Our Reply:

Thank you for the question. In the present study, we found a lower internal consistency (Cronbach’s alpha) for some of the syndrome scales, i.e., the “Withdrawn/Depressed” and the “Social Problems” scales. Cronbach’s alpha is a function of the number of items within a scale and the average correlation of the (pairs of) items. Low internal consistency may therefore partly be indicative of lack of consistency in the responses to the questions pertaining to syndrome scales. The computation of the internal consistency is based on the assumption of a reflective model for the latent variable causing the scores on the items (the level of scores on the items are reflecting the level of the latent variable). Although a reflective model is reasonable for the YSR, some of the items within a scale are not expected to be highly correlated (e.g., Social Problems: “I am jealous of others”; “I am poorly coordinated or clumsy”; “ I am too dependent on adults”). So, the latent variable social problems can be manifested in different ways in different people, and one can in such a situation think of “social problems” as a formative construct created by different symptoms of problems formed by social expectations outside the adolescents themselves. This may partly be the reason for low correlations in this case. The inconsistencies in responses on different items within a scale might be due lack of culturally appropriate items in those syndrome scales. Therefore, low scores indicate that the set of items do not reliably measure the same construct. Another reason for varying internal consistency is the varying number of items in different scales. The Withdrawn/Depressed scale has eight indicators, and this is the lowest number of items of the eight YSR syndrome scales. We have now included some of the above explanations in the Discussion section, see lines 423-433 in the revised manuscript. 

Reviewer’s comment 11: 

Line 235-236, how is the mention of the sample population to be in accordance with the overall Nepal population distribution justifiable when your sampling was purposive?

Our Reply:

Thank you for this feedback. We have now removed the sentence.

Reviewer’s comment 12:

Since YSR was not validated in Nepali and did not have the norms, I am interested to know what could have been the finding of EBP if the norms from a culturally similar country were applied!

Our Reply:

Thank you for the comment. YSR is not validated in Nepali and do not have the norms. Therefore, we have used American norms to assess adolescents’ EBPs in our study. As we have mentioned in the discussion section, according to ASEBA norms, Nepal was placed in the higher scoring country based on a previous study finding. According to findings in the present study, Nepal should be placed as a medium scoring country (https://aseba.org/societies/). As of YSR scores, the American society also belongs to the medium scoring countries. So, we have analyzed our results based on a similar scoring society. Hence, we can assume that we would have similar results even if the YSR was validated and had norms from Nepal. When searching the international literature, we did not find any studies that had established separate norms or cut-offs for EBPs in countries comparable to Nepal, e.g., in India or other South Asian countries. Hence, we will not speculate on what the results might have been had we used different, more culturally appropriate norms.

Reviewer’s comment 13:

Did you explore the relationship between EBP and two broad-band scales? It would have been interesting to know if internalizing problems were common in adolescents suffering from severe traumas.

Our Reply:

Thank you for the question. We examined if severe trauma was associated with EBPs – i.e., with Total Problems, Internalizing Problems, and Externalizing Problems, see Table 6, lines 303-304 in the revised manuscript. Negative/traumatic life events were found to be significantly associated with both Internalizing and Externalizing problems. However, we did not compare if internalizing or externalizing problems were common in adolescents suffering from severe trauma since there were few adolescents who had severe trauma. 

Reviewer’s comment 14:

Lastly, as you have explained the reasons for the increased prevalence of internalizing problems in adolescents our culture of discouraging verbal expression and emotional expression from childhood additionally might have played the role in the occurrence of internalizing problems.

Our Reply:

Thank you for the suggestion. We have elaborated some more on this - see Discussion, lines 337-338 in the revised manuscript. 

Reviewer #5

Reviewer’s comment 1: 

The Abstract clearly describes the study's aims, methods, and results. The title of the article is informative and relevant. Relevant references have been listed.

Introduction and Method:

The introduction and background list what are already known about this topic, and are clear on the research question, which is justified considering what is presently known. Subject selection is clearly stated, and the variables are defined and measured appropriately. The noted study methods are valid and reliable with the provision of enough detail to replicate the study.

Results:

Results are presented appropriately, with the provided text adding further context to the data. It is clear to the reader in terms of findings what is statistically significant and practically meaningful.

Discussion:

Results were discussed well, and placed in context, without being overinterpreted. Conclusions answer the aims of the study and are supported by results and references. The limitations of the study listed are opportunities to inform future research.

Overall:

This journal article aims to explore the prevalence of EBPs and associated adolescent factors among school-going adolescents in Nepal. It contributes to the dearth of information on this topic from an LMIC in the South Asian context and allows for the comparison of Adolescent EBPS in this region. The study design was appropriate to answer the aims, and the results do justice in adding the Nepali adolescent experience of EBPs and filling in missing information on associated factors in the country context as well as providing other relevant data on the topic. The previous reviewers’ comments have been addressed adequately and the article remains consistent with itself. The article is recommended to be accepted for publication.

Our reply:

Thank you for the encouraging comments. We appreciate your effort to go through the manuscript and giving positive feedbacks. 

Reviewer #6

Reviewer’s comment 1:

Was there any feasibility criteria used for the purposive sampling method?

Our Reply:

Thank you for the question. To clarify, we have now added the following sentence: “The purposive sampling technique was chosen for cost effectiveness and for ease of data collection and travels”, see lines 132-133 in the revised manuscript. 

Reviewer’s comment 2:

Please clarify the statement "since assent by the youths themselves was not mandatory at that time"

Our Reply:

Thank you for the comment. As mentioned in the methods section, data collection was done during 2017-2018. Existing guidelines of the Nepal Health Research Council (NHRC) at that time did not require consent taken from adolescents, only from parents. Hence, only written consent from the parents was taken in this study. However, we also obtained verbal consent from the participating adolescents. We have clarified this in the manuscript, see lines 140-141 in the revised manuscript. Later, in 2019, the NHRC guidelines were changed, and it is now mandatory in Nepal to take written assent from adolescents as well. 

Reviewer’s comment 3:

In table 1 "geographic region of residence"; in methods authors have mentioned about Kathmandu as a separate entity. Kindly clarify.

Our Reply:

Thank you for the comment. Participants of the study were selected from sixteen districts: three districts from the Mountains region, six districts from the Middle hills region, six districts from the Terai region, plus the Kathmandu district. We have now revised the text in the Methods part (“Participants and Procedure”) to show that the Kathmandu district is included in the Middle hills region, see line130, and in the Table 2 (line 242) in the revised manuscript, we have further marked that Kathmandu is included in the “Middle hills region”. 

Reviewer’s comment 4:

Table 1: Physical illness present or absent is very vague term. What physical illness were included? May be clarified in method section.

Our Reply:

Thank you for the comment. Yes, we agree that the term “physical illness” is a vague term. Reviewer 4 has commented on the same thing in comment no 8. We kindly ask you to read our reply to that comment. You will find our clarification in the Methods section, lines 190-191, and again, under limitations in the Discussion section, lines 453-456 in the revised manuscript. 

Reviewer’s comment 5:

 The claim of being the first study in Nepal would be very misleading. There has been national mental health survey in 2020 and data are available on mental health of adolescent as well.

Our reply:

Thank you for the suggestion. We agree that there are other studies, including a national mental health survey in 2020, which has reported on adolescent mental health. We have now deleted the word “first” to avoid misleading information, see line 417 in the revised manuscript in the “Strengths and limitations”, part of the Discussion. 

Reviewer’s comment 6:

 You may add some important points in discussion like the treatment gap and implication of findings to the practicing psychiatrists.

Our reply:

Thank you for the suggestions. We acknowledge that the study findings might have important implications for practicing mental health professionals. We have now added a paragraph called “Implications for mental health services” in the Discussion section, see lines 459-463 in the revised manuscript. 

Reviewer #7

Reviewer’s comments 1:

Overall comment- Pertinent, useful, important, and well-designed study; and well written manuscript, which needs to address a few issues for publication. Some issues-

• Regarding title, ‘Self-reported emotional and behavioral problems in Nepali adolescents - a general population-based study’, is it

- in Nepali adolescent students?

- A school- based study?

- As discussed by the authors themselves, whether enrolled in community or school may have implication.

Our Reply:

Thank you for the constructive feedback. Regarding the title, we agree to change it to a study among school going adolescents. We have revised the title of the study as follows: “Self-reported emotional and behavioral problems among school going adolescents in Nepal – a cross-sectional study”. 

Reviewer’s comments 2:

Concern regarding section ‘Introduction’ is only word count/limit if it is as per journal format? Or may make a succinct introduction from current one.

Our Reply:

Thank you for the comment. We have now removed some paragraphs / sentences that were not of any major importance from the introduction to shorten it. We believe the introduction as it now appears is according to the journal’s requirements. 

Reviewer’s comments 3:

Minor language and punctuation errors, e.g., better use past tense than present in ‘Materials and Method’ section (e.g., in line- 193 previous version of manuscript).

Our Reply:

Thank you for the suggestion. We have now checked the manuscript for language and punctuation errors, including use of past tense. 

Reviewer’s comments 4:

Authors wrote in discussion, ‘Consistent with some studies [24], we found that Externalizing Problems seemed to rise with increasing age.’ (line 401). Please indicate based on which result (table No.?), this has been inferred? I do not see any such result, i.e. distribution as per age groups so that we can infer it.

Our reply:

Thank you for the comment. We have this statement based on the results shown in Table 6. In the regression analysis, the regression coefficient for age and externalizing problems was positive and significant at the 0.01 level, which suggests that with increasing age, the Externalizing problems seem to rise. 

Reviewer’s comment 5: 

What was done for those adolescent students with EBPs? (An ethical concern.)

Our Reply:

Thank you for the very pertinent question. The present study is an epidemiological study, and the concern that you raised, is a major ethical concern of such studies. As for the present study, those parents and adolescents who expressed concerns about any EBPs during the data collection were offered a referral to the child and adolescent psychiatric outpatient clinic at Kanti Children’s Hospital, Kathmandu, for further evaluation and management. This offer was included in the information letter that all participating parents received together with the consent form. 

Reviewer’s comment 6: 

References need to follow the proper style consistently, e.g., in references 10, 16, 19.

Our reply:

Thank you for bringing this to our attention. We have again checked and corrected all the references to follow a proper style consistently. 

Reviewer#8

Reviewer’s comment 1:

Title: The title says the study is a general population-based study. The methodology says the study sample is school attending population. The school-attending population, however, is not the same as the general population. School students, asylum, jail inmates, etc. are special population groups that differ from the general population, and they do not represent the general population. I mean adolescents in the general population are not the same as adolescents who go to school/college. Therefore, the title of this article seems to be misleading.

Our Reply:

Thank you for this important feedback. The same comment was made by one of the other reviewers (reviewer 7). We agree that the current title of the article might be misleading. Hence, we have changed the title of the article to: “Self-reported emotional and behavioral problems among school going adolescents in Nepal - a cross-sectional study”. 

Reviewer’s comment 2:

Abstract: The first sentence of the conclusion section (of the abstract) is the repetition of what is already mentioned in the background section (of the abstract).

Our reply:

Thank you for rightly pointing this out. We have now revised the background section in the abstract to avoid repetition, see lines 25-27 in the revised manuscript. 

Reviewer’s comment 3:

Introduction: Order of information can put in better way. Initial part: What is EBPs, why is it important to know? What does the existing studies say about prevalence and correlates of EBPs and in the later part what are the lacunae in current research and why you want to do this study?

Our reply:

Thank you for your suggestion as to the order of information in the introduction part. Following your advice, we have made some changes in the organization of the text to make it more logical and coherent. Also, we have removed one smaller paragraph to shorten the introduction, see lines 57-68, 77-82 in the revised manuscript.

Reviewer’s comment 4:

Materials and methods: In section ‘Participants and procedure’: I think it is not good to tell readers to search for another article to see the detailed procedure and methodology of this study. The second sentence has the word ‘analytical sample’. I do not think ‘analytical sample’ means anything. Are YSR and TRF Nepali translations are validated?

Our reply:

Thank you for the feedback. We have now removed the sentence in the “Participant and procedure” part referring to a more detailed description of the procedure described in a previous paper. WE have given a detailed description of the procedure in lines 128-147 in the revised manuscript. We have also removed the word “analytical sample”, in line 128 in the revised manuscript. 

The main instruments used in the present study, i.e., the YSR and the TRF, have been translated into Nepali in connection with an earlier PhD study. However, the instruments were not validated in the Nepali context which is a limitation as mentioned in the “Strengths and limitations” part of the Discussion section, see lines 423-437 in the revised manuscript. 

Reviewer’s comment 5:

References: Referencing styles are not uniform.

Example:

Sahayaraj.R W. Behavioural and emotional problems in school going adolescents.

International Research journal of Management Sociology & Humanities, 2015; 6(12):357-363.

Doi: https://doi.org/10.32804/IRJMSH

In the above referencing, you can omit 3 in 363 to write it like below

Sahayaraj.R W. Behavioural and emotional problems in school going adolescents.

International Research journal of Management Sociology & Humanities, 2015; 6(12):357-63.

Doi: https://doi.org/10.32804/IRJMSH

Our reply:

Thank you for rightly pointing this out. We have now checked all the references to make sure that all the references are uniform and correct. 

Reviewer’s comment 6:

General comments:

The manuscript is better to be proofread by a professional English editor. For text clarity, we

can refrain from using additional words, mostly filler words/connecting words, which can be

omitted, e.g. "respectively", "thus", "hence", "therefore", "however, "indeed", etc. Long

sentences can be converted into more than one shorter sentence to make them more readable.

There are a few spelling mistakes. Eg Analyses of variance should be spelled correctly as

Analysis of variance (plz see abstract). Hedges’ g should be spelled as Hedges’ G. References

numbers are generally to be put at the end of the sentences. There are a few instances where

reference numbers are put in the middle of the sentences.

Our reply:

Thank you for the suggestions. Proofreading was made by a professional English editor through a professional company: “Professional Standards Editing – PSE” (pse@professionalstandardsedition.com). We have corrected the spelling mistakes that you have pointed out. Sentences where references were put in the middle, have been revised so that all references are now put at the end of the sentence.

Regarding the Hedges’ g, we decided to keep it as Hedges’ g instead of Hedges’ G since Hedges himself has written it as “g” when it is used to calculate the effect size of one experiment/observation. He used “ Hedges’ G” to give the average effect size for multiple experiments/observations.

---

## [Decision Letter · Decision Letter 2]

5 Jun 2023

Self-reported emotional and behavioral problems among school going adolescents in Nepal - a cross-sectional study

PONE-D-22-03677R2

Dear Dr. Adhikari,

We’re pleased to inform you that your manuscript has been judged scientifically suitable for publication and will be formally accepted for publication once it meets all outstanding technical requirements.

Kind regards,

Yubaraj Adhikari, Ph.D.

Academic Editor

PLOS ONE

Additional Editor Comments (optional):

Thank you for all your efforts to incorporate all the concerns of reviewers. There are still a few recommendations from the reviewers. Kindly address those comments outlined by the reviewers (#7 and 8) below in section 6. After incorporating them in the final version, this manuscript is ready to publish.

Reviewers' comments:

Reviewer's Responses to Questions

**Comments to the Author**

1. If the authors have adequately addressed your comments raised in a previous round of review and you feel that this manuscript is now acceptable for publication, you may indicate that here to bypass the “Comments to the Author” section, enter your conflict of interest statement in the “Confidential to Editor” section, and submit your "Accept" recommendation.

Reviewer #4: All comments have been addressed

Reviewer #6: All comments have been addressed

Reviewer #7: All comments have been addressed

Reviewer #8: All comments have been addressed

2. Is the manuscript technically sound, and do the data support the conclusions?

Reviewer #4: Yes

Reviewer #6: Yes

Reviewer #7: Yes

Reviewer #8: Yes

3. Has the statistical analysis been performed appropriately and rigorously? 

Reviewer #4: Yes

Reviewer #6: I Don't Know

Reviewer #7: (No Response)

Reviewer #8: I Don't Know

4. Have the authors made all data underlying the findings in their manuscript fully available?

Reviewer #4: Yes

Reviewer #6: Yes

Reviewer #7: Yes

Reviewer #8: Yes

5. Is the manuscript presented in an intelligible fashion and written in standard English?

Reviewer #4: Yes

Reviewer #6: Yes

Reviewer #7: Yes

Reviewer #8: Yes

6. Review Comments to the Author

Reviewer #4: (No Response)

Reviewer #6: (No Response)

Reviewer #7: I leave it to editorial committee to ensure word limit/ style of introduction section to the journal style.

Great work for the context of subject setting and country.

Some of the suggested punctuation and typo errors (still present) are better to be corrected.

Reviewer #8: Most of the suggestions are incorporated. Title of the research has been changed as per the study was done among school going adolescents only. Spellings and grammar mistakes are corrected. However there are still minor things which are better if corrected. Examples: Referencing styles are yet to be in uniform style. In line no 500 (reference 6), the reference text should have 'Solmi M, Radua J, Olivola M, Croce E, Soardo L, Salazar de Pablo G, et al. Age at onset of mental disorders worldwide: large-scale meta-analysis of 192 epidemiological studies. Mol Psychiatry. 2022 Jan;27(1):281-95."

Usually you ignore the web address and give the reference as you would for a printed journal article.

You should only use the electronic journal article format when the journal has no volume, issue and page numbers, or

it is not available as a print version at all (or you’re not sure), or when the article is "forthcoming”, “in press” or “online ahead of print”, so that it is available electronically but has not yet been given a place in a print issue and assigned page numbers.

You have mentioned on some occasions 'This study focuses on the prevalence, magnitude, and type of.......' For me it is not clear what you mean by 'magnitude'. If it is same as prevalence, the word 'magnitude' is not necessary to be put here.

7. PLOS authors have the option to publish the peer review history of their article (what does this mean?). If published, this will include your full peer review and any attached files.

Reviewer #4: No

Reviewer #6: **Yes: **Dr. Pawan Sharma

Reviewer #7: **Yes: **Dhana Ratna Shakya

Reviewer #8: No

---

## [Editor Report · Acceptance letter]

16 Jun 2023

PONE-D-22-03677R2 

Self-reported emotional and behavioral problems among school-going adolescents in Nepal - a cross-sectional study 

Dear Dr. Adhikari:

I'm pleased to inform you that your manuscript has been deemed suitable for publication in PLOS ONE. Congratulations! Your manuscript is now with our production department. 

Kind regards, 

on behalf of

Dr. Yubaraj Adhikari 

Academic Editor

PLOS ONE